# Slc7a5 regulates Kv1.2 channels and modifies functional outcomes of epilepsy-linked channel mutations

Victoria A. Baronas[1], Runying Y. Yang[1], Luis Carlos Morales[1], Simonetta Sipione[1,2] & Harley T. Kurata [1,2,3]

Kv1.2 is a prominent voltage-gated potassium channel that influences action potential generation and propagation in the central nervous system. We explored multi-protein complexes containing Kv1.2 using mass spectrometry followed by screening for effects on Kv1.2. We report that Slc7a5, a neutral amino acid transporter, has a profound impact on Kv1.2. Co-expression with Slc7a5 reduces total Kv1.2 protein, and dramatically hyperpolarizes the voltage-dependence of activation by −47 mV. These effects are attenuated by expression of Slc3a2, a known binding partner of Slc7a5. The profound Slc7a5-mediated current suppression is partly explained by a combination of gating effects including accelerated inactivation and a hyperpolarizing shift of channel activation, causing channels to accumulate in a non-conducting state. Two recently reported Slc7a5 mutations linked to neurodevelopmental delay exhibit a localization defect and have attenuated effects on Kv1.2. In addition, epilepsy-linked gain-of-function Kv1.2 mutants exhibit enhanced sensitivity to Slc7a5.

[1] Department of Pharmacology, University of Alberta, Edmonton T6G 2R3, Canada. [2] Neuroscience and Mental Health Institute, University of Alberta, Edmonton, Canada. [3] Alberta Diabetes Institute, University of Alberta, Edmonton, Canada. Correspondence and requests for materials should be addressed to H.T.K. (email: kurata@ualberta.ca)

Kv1.2 is a prominent voltage-gated potassium channel in the central nervous system, where it influences cellular excitability and action potential propagation[1–3]. As the first eukaryotic voltage-gated channel with a reported atomic resolution structure[4], it has been used as a template for understanding and investigating fundamental details of voltage-dependent regulation of ion channels. Genetic manipulation of Kv1.2 also illustrates important subtype-specific roles for Kv channels in the CNS. Early mouse knockout models showed a particular requirement for Kv1.2 among the Kv1 subfamily, as Kv1.2 knockout mice fail to survive beyond 3 weeks of life due to severe generalized seizures[5]. More mildly perturbative mutations of Kv1.2 have been linked to an ataxic phenotype in mice[6]. The advent of next-generation sequencing has accelerated the correlation of genetic mutations with rare phenotypes, and several Kv1.2 mutations have been identified in patients with severe epilepsies[7–10]. Molecular phenotyping of these genetic defects in heterologous systems yields basic information that may partly inform the link between the mutation and the disease, but these interpretations lack a more complete understanding of interactions between channels and extrinsic regulators, such as accessory proteins. Although our study focuses on Kv1.2, this shortcoming is likely true for many investigations of disease-linked ion channel or neurotransmitter receptor mutations.

The canonical accessory proteins for Kv1.2 and other Kv1 subtypes are Kvβ subunits, which promote cell surface maturation and (in some cases) inactivation[11–13]. Kv1.2 subunits also bind to cytoskeletal anchors, including cortactin, in a tyrosine phosphorylation dependent manner that influences Kv1.2 endocytosis[14,15]. The sigma-1 receptor is another associated protein of Kv1.2, reported to assemble with Kv1.2 and promote trafficking to the cell membrane in response to cocaine exposure[16]. Certain lipids, including phosphatidic acid, can alter the voltage-dependence of Kv1.2 activation[17]. This list of interactors is likely incomplete. For instance, several reports have described a poorly understood dynamic regulation in heterologous systems and primary dissociated neurons, generating wide cell-to-cell variability of Kv1.2 gating that likely depends on extrinsic regulatory mechanisms (not directly encoded by the primary sequence of the channel)[18,19]. Despite the variety of extrinsic factors reported to regulate Kv1.2 channel gating, none of these binding partners account for the dramatic moment-to-moment alteration of Kv1.2 activity that has been observed. Therefore, there are likely other interacting proteins and molecules with significant effects on channel gating, which have not yet been discovered.

In this study, we investigate the potential assembly of Kv1.2 with previously unrecognized accessory proteins. We use a mass spectrometry approach to identify candidate genes, followed by screening of their effects on Kv1.2. We report that Slc7a5, a neutral amino acid transporter, associates with Kv1.2 channels and dramatically alters gating and expression. Several aspects of this putative regulatory complex stand out. Slc7a5 mutations have been linked to recessively inherited neurodevelopmental delay[20], and while this has been attributed to its role as an amino acid transporter, the pleiotropy we report suggests additional mechanisms that could also contribute to severe neurological phenotypes. Second, the assembly of an ion channel and transporter is part of an emerging trend of functional interactions between complex transmembrane proteins (channels, transporters, GPCRs)[21–23]. Third, the gating effects of Slc7a5 are more dramatic than any previously reported accessory subunit of Kv1 channels, and we also describe a mechanism of current suppression involving compounded effects of accelerated inactivation and a pronounced hyperpolarizing shift of channel activation. Finally, we report that gain-of-function Kv1.2 mutations identified in patients with severe human epilepsy are particularly susceptible to suppression by Slc7a5, and this may underlie the paradoxical observation that both gain- and loss-of-function Kv1.2 mutations lead to severe epilepsy[10].

## Results

**Identification of Kv1.2-associated proteins**. We have previously reported marked variability of Kv1.2 gating parameters in commonly used expression systems, suggesting these channels are subject to unrecognized regulatory mechanisms[18,24]. To identify interacting proteins, we used 1D4 affinity purification of cross-linked channel complexes, followed by quantitative LC–MS/MS mass spectrometry. Cross-linking and mass spectrometry was done using a hybrid channel (Kv1.5N/Kv1.2[S371T]) to enhance cell surface maturation in HEK293 cells, followed by functional validation using wild-type Kv1.2 (rat). Several previously reported Kv1.2 regulatory proteins appeared in the screen (data supplement), including several phosphatases and kinases[14,25], and a RhoA guanine nucleotide exchange factor[26]. Based on criteria including abundance relative to pulldowns from untransfected cells, and cross-referencing against the CRAPome[27] we selected 30 candidate proteins for further screening by electrophysiology in *ltk-* fibroblasts (Fig. 1a, a full list of proteins identified has been deposited online, see Data Availability statement). Our previous study highlighted a prominent effect of redox conditions on Kv1.2[24], so we also purified protein in either ambient redox or reducing conditions, and prioritized abundant proteins previously identified to contain labile extracellular disulfide bonds[28] (Fig. 1a). Electrophysiology recordings were performed in ambient redox conditions (Fig. 1b, c) and we measured the half-activation voltage ($V_{1/2}$, Fig. 1b) and current density (Fig. 1c). Most proteins tested had little or no discernible impact on Kv1.2 electrical function, although we cannot rule out that they may affect Kv1.2 under other experimental conditions. The neutral amino acid transporter Slc7a5 (LAT1) had the most pronounced impact on Kv1.2, greatly reducing Kv1.2 currents and shifting the conductance-voltage relationship by approximately −40 mV (Fig. 1b, c) in our initial screen.

**Suppression of Kv1.2 currents by Slc7a5**. We further explored the effects of Slc7a5 on Kv1.2 using electrophysiology (Fig. 2a) and western blot (Fig. 2b–d). Co-expression of Kv1.2 and Slc7a5 (2:1 transfection ratio) markedly decreased Kv1.2 current density, recapitulating our initial screen (using a holding potential of −80 mV, and a voltage step to +60 mV). We also tested Slc3a2, which also appeared with high abundance in the screen (Fig. 1a) and can form a heterodimer with Slc7a5[29]. Slc3a2 and Slc7a5 peptides were both enriched in Kv1.2 cross-linked complexes, but Slc7a5 was not well identified by mass spectrometry (2 peptides, small sequence coverage), likely because it is primarily composed of transmembrane segments. Co-expression with Slc3a2 alone did not affect Kv1.2 current density. The combination of Kv1.2, Slc3a2, and Slc7a5 (2:2:1 transfection ratio) partially rescued the Slc7a5-mediated suppression of Kv1.2 currents (Fig. 2a). These experimental conditions were chosen to highlight the broad range of regulation that is possible with various combinations of Kv1.2, Slc7a5, and Slc3a2. Although, we have not examined this in detail, our preliminary exploration demonstrated that co-transfection with larger amounts of Slc7a5 (e.g., a 1:1 ratio with Kv1.2) caused less prominent rescue by Slc3a2. Overall, these data suggest that Kv1.2 function is influenced by the relative amounts of Slc7a5, Slc3a2, and Kv1.2.

We also measured Kv1.2 protein expression after transient transfection with Kv1.2 and combinations of Slc7a5 and Slc3a2 (Fig. 2b–d). Kv1.2 generates two prominent bands: a mature cell

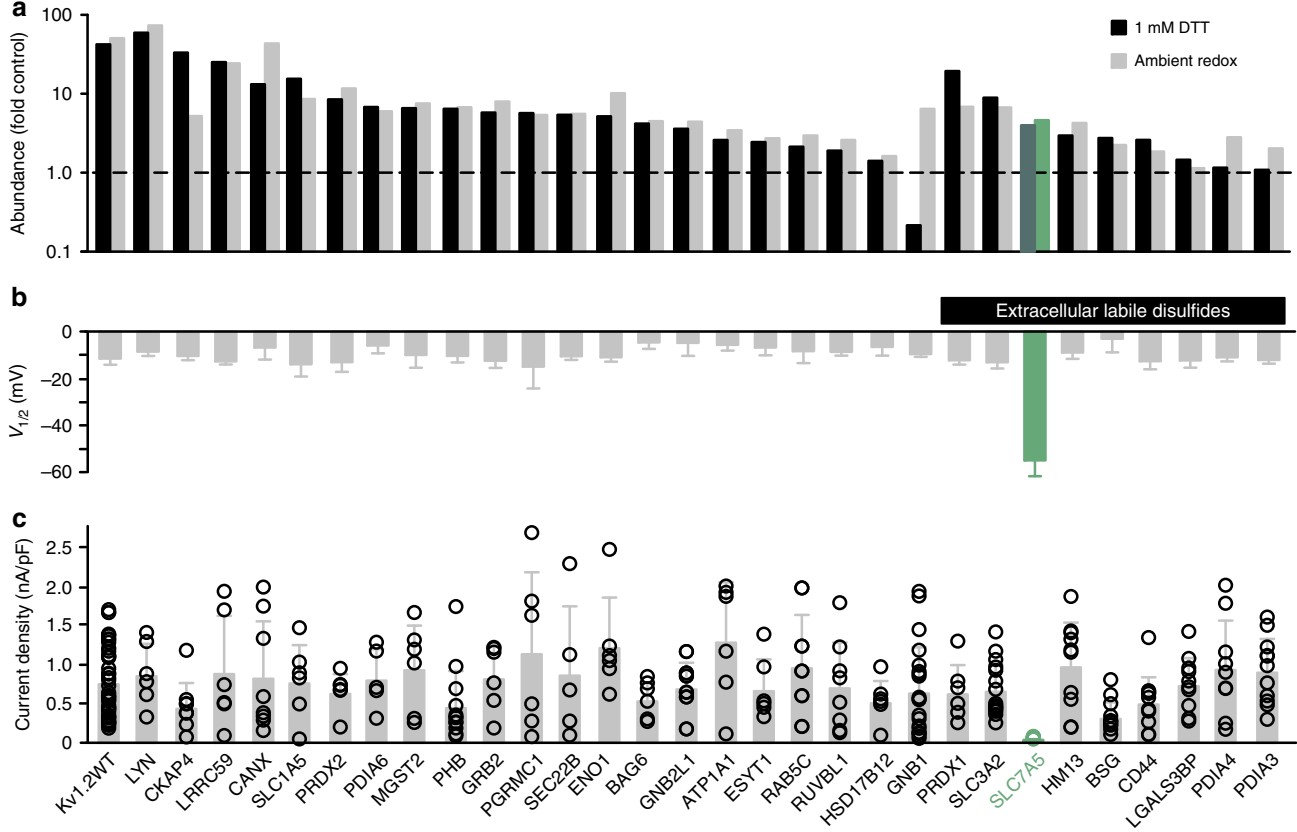

**Fig. 1** Mass spectrometry and screening of Kv1.2 channel complexes. Kv1.2-1D4 was expressed in HEK cells, and EGS-cross-linked complexes were immunoprecipitated and analyzed by LC–MS/MS. Thirty proteins were chosen for further screening. **a** Abundance of proteins relative to untransfected cells, with cross-linking performed in ambient redox conditions (gray) or 1 mM DTT (black). Proteins containing labile extracellular disulfides are indicated by the black bar. **b** $V_{1/2}$ (mean ± s.d.) was determined for Kv1.2 when co-expressed with each candidate gene (mEGFP-tagged) in *ltk*-mouse fibroblasts. Conductance-voltage relationships were measured by stepping between −120 and +60 mV (100 ms in 10 mV steps, −80 mV holding potential) followed by a tail current voltage of −30 mV. In addition, a 100 ms depolarization to +60 mV was delivered before each sweep to relieve use-dependent activation ($n = 5$-15, $n = 31$ for WT Kv1.2). **c** Kv1.2 current densities at +60 mV (pulsed from a holding potential of −80 mV) after co-expression with each candidate gene. The data from individual cells are superimposed on bars that represent mean ± s.d. ($n = 5$-15, $n = 31$ for WT Kv1.2)

surface band, and a core glycosylated band[30]. Slc7a5 + Kv1.2 (1:1) co-expression diminished total Kv1.2 expression by 50 ± 15% (mean ± s.d., here and throughout the text), relative to Kv1.2 alone. Slc3a2 + Kv1.2 (1:1) did not significantly affect overall expression (116 ± 43%), and co-expression of Slc3a2 with Slc7a5 + Kv1.2 (1:1:1) partially rescued Kv1.2 expression, to 68 ± 3% (Fig. 2c), although this was not statistically significant compared to WT Kv1.2 expression. In all cases, surface expression as a fraction of total protein was not affected (Fig. 2d).

We noted an inconsistency between Kv1.2 current density vs. protein expression, depending on the expression of Slc7a5 and Slc3a2. For example, co-expression of Kv1.2 + Slc7a5 + Slc3a2 did not dramatically rescue protein expression (relative to Kv1.2 + Slc7a5, Fig. 2b–d), but had consistently larger currents (Fig. 2a). Thus, it was unclear why Slc7a5 caused more pronounced current suppression. We recognized that Slc7a5-mediated current suppression is likely a combination of effects on expression and gating, because hyperpolarized holding voltages result in significant disinhibition of Kv1.2 co-expressed with Slc7a5 (Fig. 3a). We measured Kv1.2 current at 0.3 s intervals, with holding potentials between −80 mV and −120 mV. We observed that with a holding potential of −120 mV, Kv1.2 currents increase significantly during a 30 s pulse train, but not with −80 or −100 mV (Fig. 3b, c). We are uncertain whether significantly more disinhibition can be achieved with more negative holding

potentials or a longer time at −120 mV, as the long application of strong hyperpolarized voltages was a technical challenge that frequently led to seal breakdown. As described later, it is also not clear whether this disinhibition reflects recovery from a canonical C-type inactivated state, or some other non-conducting state.

**Slc7a5 induces a hyperpolarizing shift of Kv1.2 activation.** The recognition of hyperpolarization-mediated disinhibition allowed us to examine gating effects in more detail. Kv1.2 + Slc7a5 co-expression (2:1 transfection ratio) leads to a hyperpolarized $V_{1/2}$ of Kv1.2 activation of −58 ± 3 mV, compared to −11 ± 3 mV in Kv1.2 channels expressed alone. Co-expression of Slc3a2 with Kv1.2 (1:1 transfection ratio) does not affect the $V_{1/2}$ (−11 ± 10 mV). However, similar to the effects on current density, co-expression of Kv1.2 with both Slc7a5 and Slc3a2 (2:1:2 ratio of Kv1.2:Slc7a5:Slc3a2) rescues the Slc7a5-mediated gating shift, generating a $V_{1/2}$ of −16 ± 3 mV (Fig. 4a, b). Slc3a2 and Slc7a5 interact via an extracellular disulfide bond between Slc7a5 Cys164 and Slc3a2 Cys109, along with other intermolecular contacts[31]. The disulfide bond is not essential for their mutual regulation of Kv1.2, as the Slc7a5[C164A] mutant transporter retains the ability to shift Kv1.2 gating, and causes a sevenfold current disinhibition at negative voltages (Supplementary Figure 1a, b). Also, gating effects mediated by Slc7a5[C164A] are rescued by

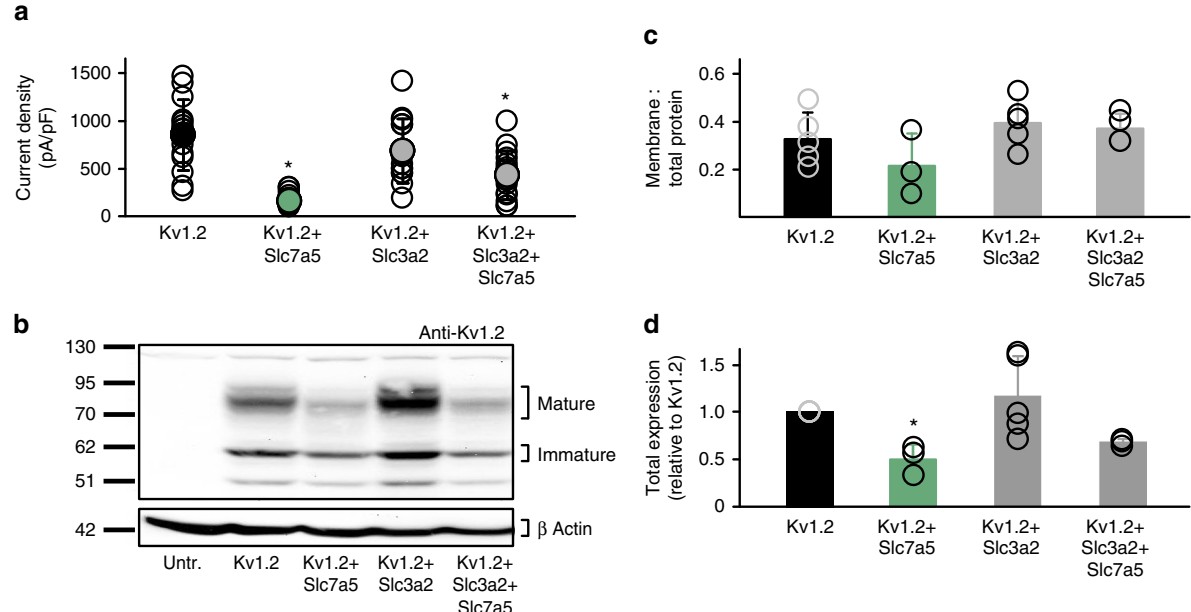

**Fig. 2** Slc7a5 alters Kv1.2 expression and current density. **a** Combinations of Kv1.2, mCherry-Slc7a5, and mEGFP-Slc3a2 were expressed in *ltk*-mouse fibroblasts. Current density was measured at +60 mV (pulsed from a holding potential of −80 mV). Transfection ratios were (with Kv1.2 maintained constant): Kv1.2:Slc7a5 (2:1); Kv1.2:Slc3a2 (1:1); Kv1.2:Slc7a5:Slc3a2 (2:1:2), $n = 14$–19 cells. A non-parametric Kruskal–Wallis ANOVA on ranks test, followed by Dunn's post hoc was used to compare treatments to WT Kv1.2, (* indicates $p < 0.05$ relative to WT Kv1.2 current density). **b** Exemplar anti-Kv1.2 western blot of *ltk*-mouse fibroblasts transfected as in **a** except Kv1.2 + Slc7a5+Slc3a2 (1:1:1) for 72 h. **c** Densitometry measurements of the cell surface:total Kv1.2 protein. No significant differences were detected between groups ($n = 3$–5). **d** Densitometry of total Kv1.2 protein, normalized to WT Kv1.2 alone. ANOVA followed by Dunnett's post hoc test was used to compare treatments to WT Kv1.2 (* indicates $p < 0.05$ relative to WT Kv1.2 transfection). The data from individual experiments are superimposed on bars that depict mean ± S.D. ($n = 3$–5)

overexpression of Slc3a2 (Supplementary Figure 1a, c), although the extent of rescue seen in co-expression with Slc3a2 varies on a cell-to-cell basis (gray lines, Supplementary Figure 1a). Thus, the Slc7a5:Slc3a2 interaction does not rely solely on the formation of an extracellular disulfide bond, consistent with a previous report describing a broad interaction surface between Slc7a8 and Slc3a2[32].

We have not exhaustively tested the entire Slc7 transporter family. However, the effects of Slc7a5 exhibit some specificity, as Kv1.2 is not affected by co-expression with Slc7a6, a closely related transporter that also heterodimerizes with Slc3a2[33]. Also, the Slc1a5 amino acid transporter does not influence Kv1.2 gating or current magnitude (Supplementary Figure 2a). Finally, Slc7a5 exhibits some degree of subtype specificity, as co-expression of Slc7a5 did not influence Kv1.5 (Supplementary Figure 2b). Future testing of other Kv1 channel types and Slc7 transporters will catalog the specificity of reported effects.

The canonical accessory subunits of Kv1 family channels are the Kvβ subunits. We tested whether Slc7a5 regulates channels co-expressed with Kvβ1.3, which introduces rapid N-type inactivation[11,12]. Co-expression with Kvβ1.3 does not prevent Slc7a5-mediated gating or disinhibition effects (Fig. 4c, d). In addition, rapid inactivation was apparent at positive voltages even when Kvβ1.3 was co-expressed with Slc7a5 (Fig. 4e). This combination of rapid inactivation together with a pronounced activation shift demonstrates that channels can simultaneously co-assemble with both Slc7a5 and Kvβ subunits.

**Slc7a5 promotes Kv1.2 inactivation.** The hyperpolarization-induced current disinhibition led us to speculate that Slc7a5 might influence Kv1.2 inactivation. To investigate this, we used the Kv1.2[V381T] mutant, which replaces the outer pore residue equivalent to *Shaker* Thr449, making channels more prone to C-

type inactivation[34]. Kv1.2[V381T] channels exhibit a similar Slc7a5-mediated gating shift as WT Kv1.2 (Fig. 5a), along with disinhibition at −120 mV (Fig. 5b). Also, Slc7a5 co-expression markedly accelerates the rate of inactivation of Kv1.2[V381T] (Fig. 5c) and shifts the steady-state inactivation curve from −31 ± 3 mV to −69 ± 6 mV (Fig. 5d). There is an important mismatch that should be noted between inactivation (Fig. 5d) and disinhibition (Fig. 5b). Disinhibition at −120 mV (Fig. 5b) was always measured before the inactivation protocol. However, subsequent recording of inactivation did not restore the original state/level of current inhibition, as currents measured in the inactivation protocol recovered with an interpulse holding voltage of −100 mV (7 s duration). These findings suggest that Slc7a5 promotes C-type inactivation of Kv1.2[V381T] and may also promote additional inactivated/non-conducting states over longer periods of time (i.e., during incubation time prior to recording).

We also tested Slc7a5 effects on the Kv1.2[I304L][S308T] (Kv1.2[LT]) mutant, which shifts channel activation to depolarized voltages by dissociating voltage sensor movement from channel opening[35]. We hypothesized that Kv1.2[LT] would be less prone to current suppression by Slc7a5 because fewer channels would activate at resting voltages. Slc7a5 caused a hyperpolarizing shift of activation of Kv1.2[LT] (Fig. 5e). More importantly, hyperpolarization to −120 mV did not cause disinhibition, suggesting that Kv1.2[LT] channels have not accumulated in an inactivated/non-conducting state (Fig. 5f, g). Taken together, these findings suggest that Slc7a5 promotes C-type inactivation (coupled to channel opening) and likely other non-conducting states of Kv1.2, partially contributing to the Slc7a5-mediated current suppression (Fig. 2a). In combination with the shift in voltage dependence of channel activation, this effect leads to a requirement for strong hyperpolarizing voltages for disinhibition of current (Fig. 3).

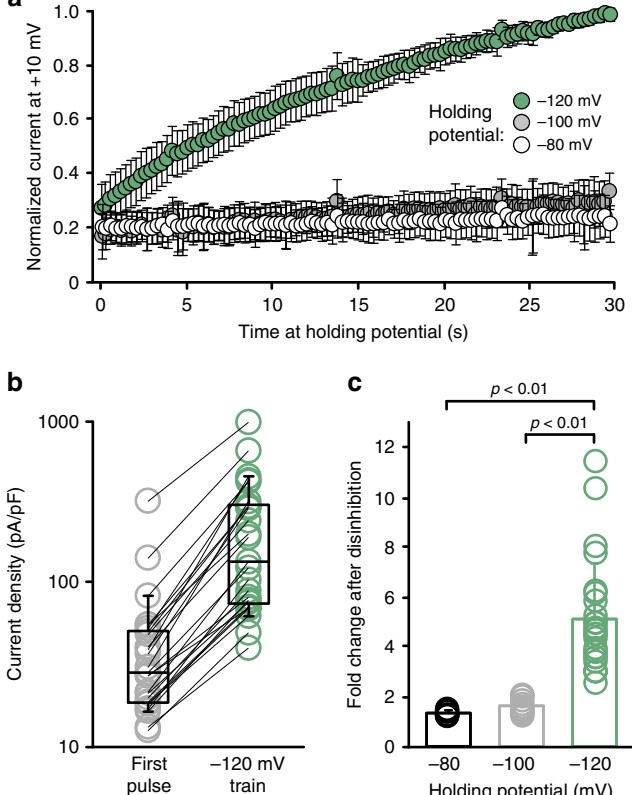

**Fig. 3** Hyperpolarization disinhibits Kv1.2 co-expressed with Slc7a5. **a** *ltk*-mouse fibroblast cells transfected with Kv1.2 + Slc7a5 (1:1) were held at a range of voltages: −80 mV ($n = 7$), −100 mV ($n = 12$) and −120 mV ($n = 25$). Currents were measured at +10 mV (50 ms pulses, every 300 ms), and normalized to the peak current after 30 s at −120 mV (mean ± s.d.). **b** Cell-by-cell current density (at +10 mV) of the first (gray) and final (green) pulses of a train of depolarizations with a −120 mV holding voltage. **c** Fold change between the initial and final test pulses of a 30 s pulse train with the indicated holding voltages (mean ± s.d.: 1.4 ± 0.1 at −80 mV; 1.6 ± 0.3 at −100 mV; 5 ± 2-fold at −120 mV, ANOVA followed by Dunnett's post hoc test was used to compare groups)

**Kv1.2 and Slc7a5 are in close physical proximity**. We further investigated the physical nature of the Kv1.2:Slc7a5 interaction using a bioluminescence resonance energy transfer (BRET) approach to assess their proximity in HEK293 cells. We fused the Nanoluc bioluminescent donor to Kv1.2 and used mEGFP as an acceptor in various test constructs. We collected emission spectra between 400 and 700 nm, resulting in a large emission peak centered at ~455 nm corresponding to Nanoluc bioluminescence, and variable levels of a shoulder with a peak centered at ~510 nm corresponding to mEGFP emission. Emission spectra were well fit by the sum of weighted components of the Nanoluc and mEGFP spectra (Fig. 6a).

We tested a variety of mEGFP-tagged BRET acceptors. Kv1.2 can assemble as a homotetramer, therefore co-expression with mEGFP-Kv1.2 generates mEGFP emission. Although not quite as pronounced, mEGFP-Slc7a5 generated a clearly discernable emission, whereas the Slc1a5 negative control generated a much smaller emission (Fig. 6a, b). We calculated the area under the curve (AUC) of the mEGFP emission component between 480 and 600 nm and normalized each AUC to a matched mEGFP-Kv1.2 positive control (run in parallel with each experiment, Fig. 6c). We also tested whether co-expression of Slc3a2 would influence the BRET signal from mEGFP-Slc7a5. In the presence

of Slc3a2, the BRET signal from mEGFP-Slc7a5 was modestly attenuated although these results are variable and ambiguous (Fig. 6c, co-expression with Slc3a2 is not significantly different from EGFP-Slc7a5 alone, but also not different from EGFP-Slc1a5). These, along with our functional findings (Figs. 2, 4), suggest that Kv1.2 and Slc7a5 are in close proximity and that Slc3a2 may influence this association, although it is not clear whether Slc3a2 occludes the Slc7a5:Kv1.2 interaction or acts by some other mechanism. Data further investigating mutual interactions of Kv1.2-Slc7a5-Slc3a2 is presented later.

We also investigated the presence of Slc7a5 in neurons. The predominant focus of investigation of Slc7a5 in the brain has been its role in vascular endothelium as a component of the blood–brain barrier[36]. Early immunohistochemical studies demonstrate enrichment of Slc7a5 protein in the blood–brain barrier, although these studies remark on the presence of a small amount of Slc7a5 throughout the brain and an enrichment in certain regions including dentate gyrus[37]. This is consistent with more recent RNA-seq approaches that detect high levels of Slc7a5 mRNA in vascular endothelium, and lower levels Slc7a5 in single neurons or enriched populations of neurons, comparable to Kv1.2 and other prominent neuronal Kv1 channel subunits, Kv1.1 and Kv1.4[38,39] (online resources including https://web.stanford.edu/group/barres_lab/brain_rnaseq.html; mousebrain.org; neuroseq.janelia.org). Consistent with these reports, we detected Slc7a5 protein (Fig. 6d) and mRNA (Fig. 6e) in cytosine arabinoside-enriched cultures of cortical neurons. We also detected Slc7a5 and Kv1.2 by immunohistochemistry in dissociated hippocampal or cortical neurons (Fig. 6f, additional images with neuronal markers are in Supplementary Figure 3).

**Mutual interactions of Kv1.2 and Slc7a5 and Slc3a2**. We further investigated interactions between Kv1.2, Slc7a5, and Slc3a2, using a flow cytometry approach to determine whether expression of one protein might influence assembly of the other two. We used complementation of split YFP fragments (YFPC or YFPN) fused to two subunits as a crude assessment of protein interaction, and tested whether the third subunit could influence YFP reconstitution[40]. For example, we tested whether YFP reconstitution between Kv1.2-YFPC and Slc7a5-YFPN was altered by expression of Slc3a2-LSS-mOrange (or LSS-mOrange alone as a control). For all permutations tested, we found only modest effects of the LSS-mOrange tagged subunit on reconstitution of YFP (Fig. 7a), suggesting that none of the subunits dramatically prevent assembly of the other two (Fig. 7a). For example, in the representative flow cytometry experiment in Fig. 7b, cells with the brightest LSS-mOrange fluorescence (from Slc3a2) also tended to have the brightest YFP signal (from the assembly of Kv1.2-YFPC and Slc7a5-YFPN), suggesting that Slc3a2 does not prevent assembly of Slc7a5 and Kv1.2.

An important aspect of the flow cytometry experiments is that split-YFP complementation can promote/stabilize the targeted protein interaction. We investigated these constructs in greater detail using patch clamp recordings of various combinations of tagged Kv1.2, Slc7a5, and Slc3a2 (Fig. 5c–e) in order to test the mutual regulation of these proteins when assembly of specific pairs is biased by split-YFP assembly. For example, we tested Slc3a2 effects when assembly of the Kv1.2:Slc7a5 complex was enhanced using Kv1.2-YFPC and Slc7a5-YFPN constructs. In these experiments, co-expression of Kv1.2-YFPC and Slc7a5-YFPN mimicked the gating shift and current suppression observed with WT constructs (Fig. 7c, d). Co-expression with Slc3a2-LSS-mOrange generated a wide range of gating phenotypes despite the irreversible fusion of Slc7a5 and Kv1.2,

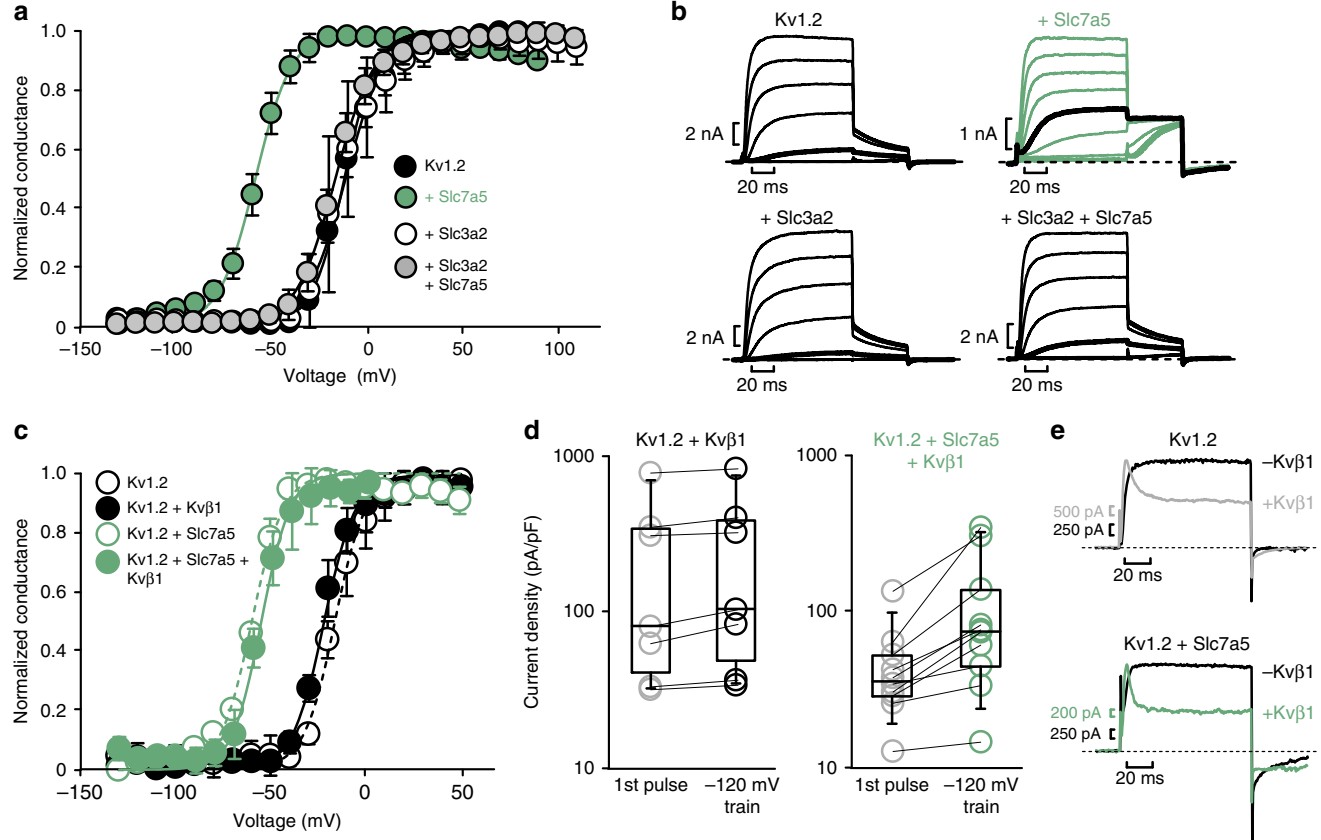

**Fig. 4** Slc7a5 shifts the voltage-dependence of Kv1.2 activation. **a** Indicated combinations of Kv1.2, Slc7a5, and Slc3a2 were transfected in *ltk-* mouse fibroblasts. Cells were hyperpolarized to −120 mV for 30 s prior to recording, leading to disinhibition of Kv1.2 currents. Conductance-voltage relationships were measured as described in Fig. 1. Conductance-voltage relationships were generated from the tail current amplitudes and fit with a Boltzmann function (Kv1.2 $V_{1/2}$ = −11 ± 3, $k$ = 11 ± 3 mV, $n$ = 5; Kv1.2 + Slc7a5 $V_{1/2}$ = −58 ± 3, $k$ = 10 ± 2 mV, $n$ = 16; Kv1.2 + Slc3a2 $V_{1/2}$ = −11 ± 10, $k$ = 10 ± 3 mV, $n$ = 10; and Kv1.2 + Slc7a5 + Slc3a2 $V_{1/2}$ = −16 ± 3 mV, $k$ = 11 ± 2 mV, $n$ = 9). **b** Representative traces of the activation curves measured in **a**, with the pulse to −20 mV highlighted. **c** HEK cells were transfected with equal amounts of the indicated combinations of Kv1.2, Slc7a5, and Kvβ1.3. Conductance-voltage relationships were generated as in **a** (Kv1.2 $V_{1/2}$ = −16 ± 3, $k$ = 9 ± 4 mV; Kv1.2 + Kvβ $V_{1/2}$ = −22 ± 4, $k$ = 9 ± 2 mV; Kv1.2 + Slc7a5 $V_{1/2}$ = −60 ± 1, $k$ = 9 ± 1 mV; and Kv1.2 + Slc7a5 + Kvβ $V_{1/2}$ = −56 ± 3 mV, $k$ = 9 ± 3 mV). **d** Current density at +10 mV was measured at the beginning and of a pulse train with a −120 mV holding potential (identical to Fig. 3; Kv1.2 + Kvβ $n$ = 7; Kv1.2 + Kvβ + Slc7a5 $n$ = 11). **e** Representative traces illustrating N-type inactivation conferred by Kvβ in the presence and absence of Slc7a5 (100 ms step from −80 mV to +60 mV)

suggesting that Slc3a2 can alter the effects of Slc7a5 on Kv1.2 even when the channel and transporter are held in close proximity (Fig. 7c, d). A similar experiment was done using Slc7a5-YFPC and YFPN-Slc3a2, along with LSS-mOrange-Kv1.2, suggesting that close association of Slc3a2 and Slc7a5 does not prevent regulation of Kv1.2 (Fig. 7e). Interestingly, in this instance, the YFP-stabilized association of Slc3a2 and Slc7a5 is not very effective at rescuing the WT Kv1.2 $V_{1/2}$ (most cells were only poorly rescued, multiple gray lines in Fig. 7e represent GV curves from individual cells). This observation may arise because Slc3a2 association helps to stabilize Slc7a5 at the membrane where it can influence Kv1.2. Also, in these experiments a 1:1:1 transfection ratio was used, possibly leading to a greater amount of Slc7a5 relative to Kv1.2, and thus weaker rescue by Slc3a2 as mentioned earlier. Overall, these experiments demonstrate complex mutual regulation of Kv1.2, Slc7a5, and Slc3a2, that persists even when strategies are taken to bias the interaction of specific interacting pairs. This finding supports the proposal that Kv1.2, Slc7a5, and Slc3a2 may form a complex, and that the rescue effect of Slc3a2 does not require physical occlusion of the Kv1.2:Slc7a5 interaction. However, the variability of channel gating observed when all three proteins are co-expressed suggests additional complexity that we cannot yet control.

**Attenuated effects of disease-linked Slc7a5 mutants**. Slc7a5 mutations that impair amino acid transport have been linked to recessively inherited forms of autism spectrum disorder[20]. We tested the effects of these disease-linked mutants on Kv1.2 gating and expression. Slc7a5[A246V] had no measurable effect on Kv1.2 gating, current density, or disinhibition at −120 mV (Fig. 8a–c, dark blue). Co-expression of Slc7a5[P375L] had attenuated gating effects relative to WT Slc7a5, and also some suppression of current expression (Fig. 8a–c, light blue). Neither Slc7a5 mutant generated a BRET signal above the Slc1a5 negative control when co-expressed with Kv1.2-Nanoluc (Fig. 8b). We regularly used western blots to confirm that these mutants are expressed at comparable levels to WT Slc7a5 (all clones were mCherry-tagged, Fig. 8d, so they run at a different molecular weight than endogenous Slc7a5 in Fig. 6d). The Slc7a5 mutants may have a variety of defects that account for their weak effects on Kv1.2. Previous reports in a reconstituted liposome assay have suggested that the mutations diminish transport activity[20]. We have noticed that these mutations also have a localization defect. Images from dissociated cortical neurons nucleofected with various Slc7a5 mutants demonstrate accumulation in intracellular puncta, in contrast to the more uniform distribution of WT Slc7a5 (Fig. 8e–g, additional images in Supplementary Figure 4).

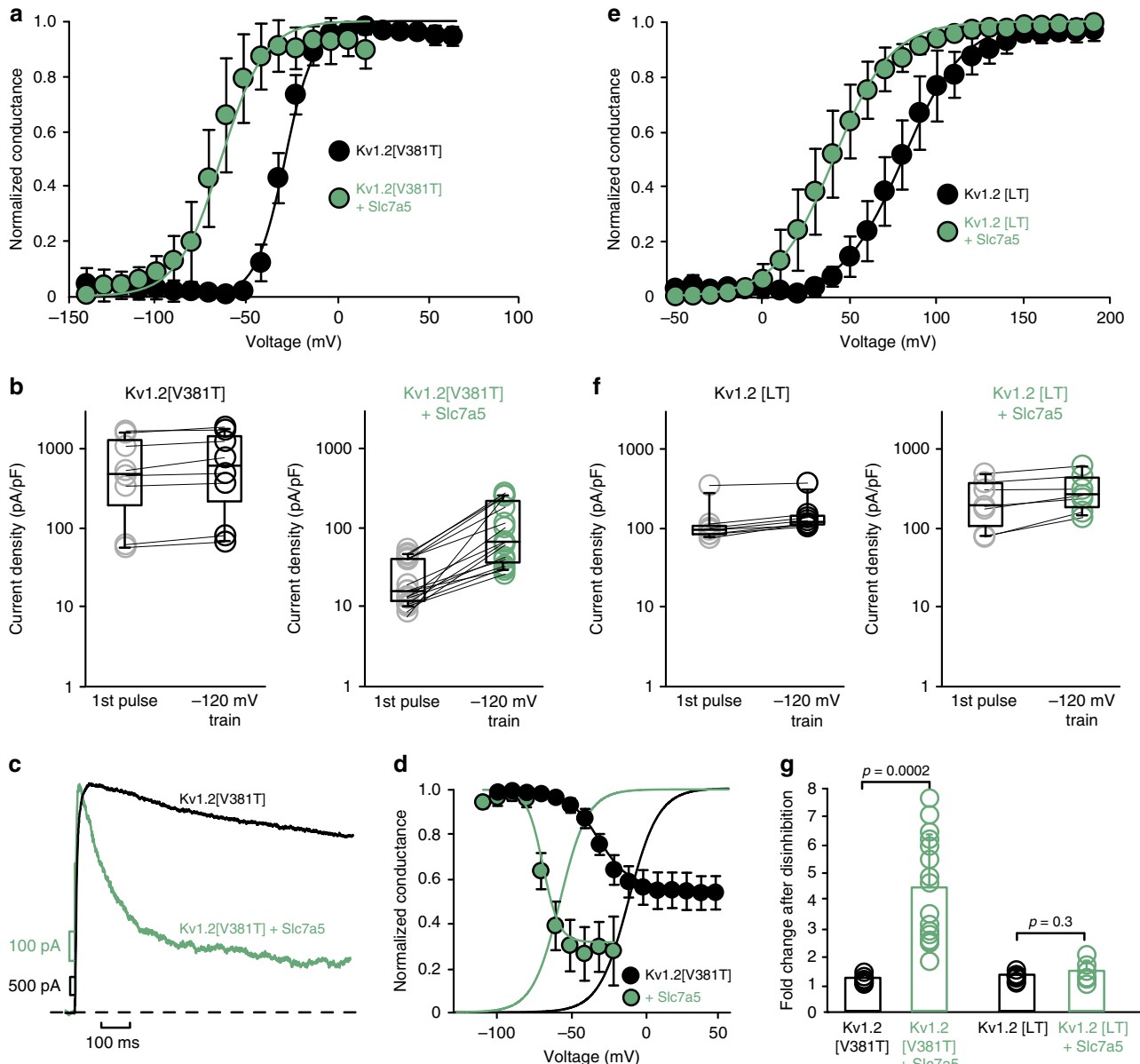

**Fig. 5** Slc7a5 influences inactivation of Kv1.2. **a** Kv1.2[V381T] channels were expressed alone or with Slc7a5 (1:1 transfection ratio). Conductance-voltage relationships were generated with the same protocol as Fig. 4, after disinhibition by a 30 s hyperpolarization to −120 mV (Kv1.2[V381T] $V_{1/2} = -16 \pm 4$ mV, $k = 8.5 \pm 0.7$ mV, $n = 6$; + Slc7a5 $V_{1/2} = -55 \pm 11$ mV, $k = 13.1 \pm 3.2$ mV, $n = 8$). **b** Current density at +60 mV was measured at the beginning and of a pulse train with −120 mV holding potential (identical to Fig. 3; Kv1.2[V381T] $n = 8$; +Slc7a5 $n = 16$). **c** Exemplar traces of Kv1.2[V381T] ± Slc7a5 (green) currents elicited by a 1 s depolarization to +60 mV (after current disinhibition by holding at −120 mV). **d** Steady-state inactivation was measured by pulsing to test voltages for 6 s and allowing channels to recover for 7 s at −100 mV. Curves were fit with a Boltzmann function (Kv1.2[V381T] $V_{1/2} = -31 \pm 3$ mV, $k = 11.1 \pm 2.1$ mV, $n = 5$; Kv1.2[V381T] + Slc7a5 $V_{1/2} = -69 \pm 6$ mV, $k = 6.7 \pm 1.8$ mV, $n = 5$). **e**, **f** Identical experiments as in **a**, **b** were performed with Kv1.2[LT] channels (Kv1.2[LT] $V_{1/2} = +80 \pm 10$ mV, $k = 18 \pm 2$ mV, $n = 7$; +Slc7a5 $V_{1/2} = +40 \pm 11$ mV, $k = 17 \pm 2$ mV, $n = 7$). **g** Current disinhibition (fold change in current after a −120 mV pulse train) for Kv1.2[V381T] ($n = 8, 16$) or Kv1.2[LT] ($n = 7, 7$). Bars represent mean ± s.d., Student's $t$-test was used to compare Slc7a5-mediated disinhibition vs. control with each channel type

**Disease-linked Kv1.2 mutants are hypersensitive to Slc7a5.** We tested the effect of Slc7a5 on two recently reported disease-linked mutations in Kv1.2, Arg297Gln and Leu289Phe[10]. When expressed alone, these mutations cause a gain-of-function involving a hyperpolarizing shift of the conductance-voltage relationship relative to WT Kv1.2 (Fig. 9c–f), and modest or absent disinhibition after hyperpolarization to −120 mV (Fig. 9a, b). However, both mutants are extremely susceptible to Slc7a5, leading to dramatic current reduction, and hyperpolarizing shifts of channel activation (Kv1.2[R297Q] $V_{1/2} = -143 \pm 6$ mV; Kv1.2

[L298F] $V_{1/2} < -200$ mV; Fig. 9a–f). This is especially evident in the exemplar traces where current inhibition coupled with the shift in activation can be better appreciated (Fig. 9c, d). The large shift in voltage-dependent activation provides a rationale for the low current density and very weak extent of disinhibition observed at −120 mV (Fig. 9a, b). Since the activation curve of the mutant channels is so dramatically shifted, significantly more negative voltages may be required for disinhibition. Using western blots, we demonstrated that expression of the mutant channels was slightly reduced relative to WT Kv1.2 channels and

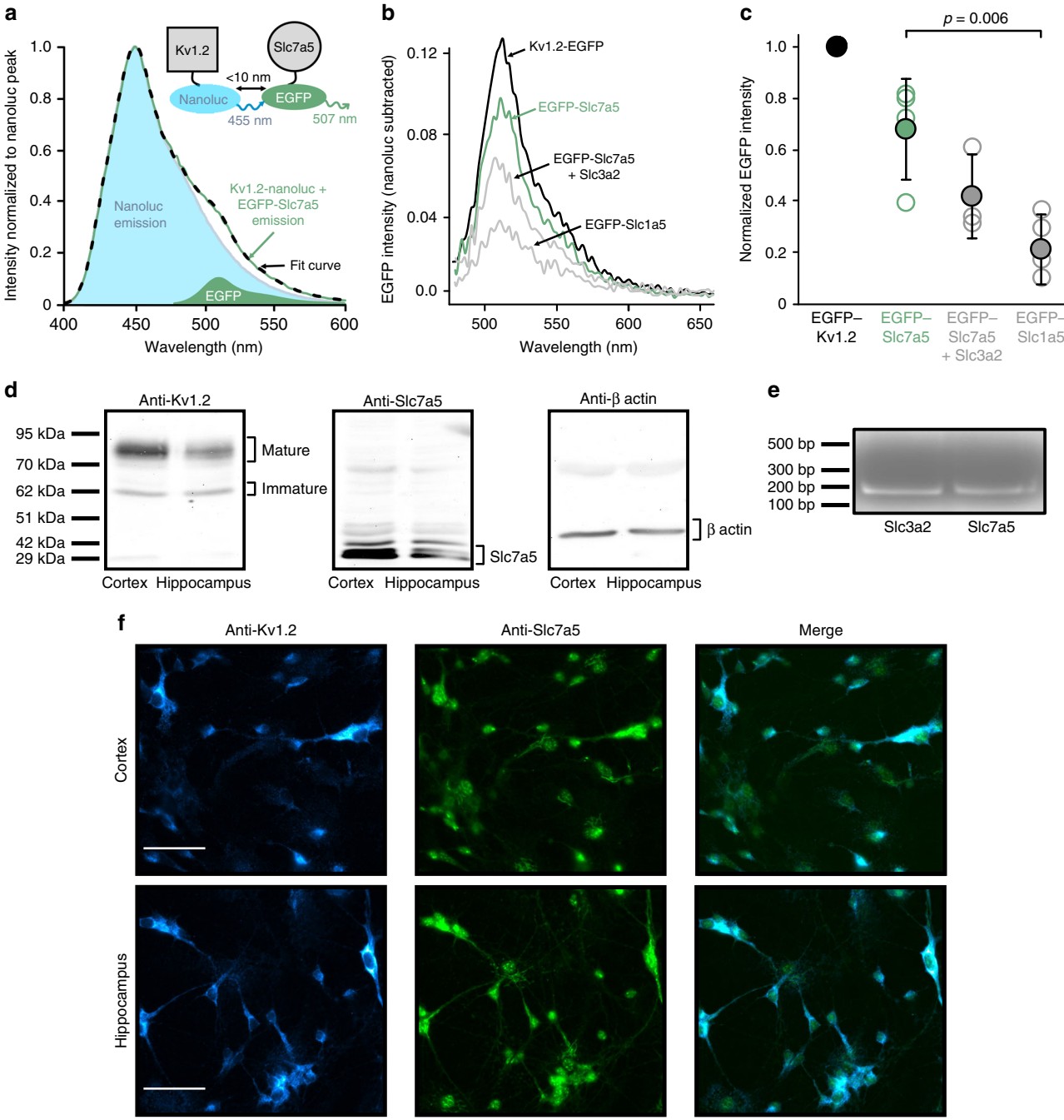

**Fig. 6** Measurement of the proximity of Slc7a5 and Kv1.2 with bioluminescence resonance energy transfer (BRET). **a** Emission spectra were collected from HEK293 cells transfected with Kv1.2-nanoluc + EGFP-Slc7a5 (green line normalized to peak). The Kv1.2-nanoluc (donor) spectrum was subtracted to yield the mEGFP component. Weighted components of the nanoluc (1.0) and mEGFP (0.1) spectra were used to fit the experimental spectrum (black dashed line). **b** mEGFP fluorescence (nanoluc-subtracted) was measured for Kv1.2-nanoluc co-expressed with various acceptor constructs, as indicated. **c** The area under the curve (AUC) for each BRET acceptor in **b** was normalized to the positive control AUC (Kv1.2-EGFP). Data are shown as mean ± SD ($n = 4$–5). ANOVA and a Tukey post hoc test were used to compare the BRET signals from EGFP-Slc7a5 (±Slc3a2) and EGFP-Slc1a5. **d** Western blot of dissociated cortical and hippocampal neurons from P2 rat pups at 14 days in vitro blotted with anti-Kv1.2 (left), anti-Slc7a5 (middle) and anti-β actin (right). **e** RT-PCR of Slc7a5 and Slc3a2 from dissociated cortical neurons from P1 rat pups after 4 days in vitro. **f** Cortical and hippocampal neurons from P2 rat pups at 7 days in vitro fixed with 4% paraformaldehyde, permeabilized with 1% Triton X-100 and stained with anti-Kv1.2 and anti-Slc7a5 primary antibodies as in **d** and fluorescent secondary antibodies. Images are representative of three neuronal cultures. Scale bars represent 50 μM

decreased further by co-expression of Slc7a5 (Fig. 9g, h). Overall, these disease-linked Kv1.2 mutants categorized as gain-of-function appear to be strongly suppressed by Slc7a5 due to diminished protein expression and hypersensitivity to the gating effects of Slc7a5.

Interestingly, the enhanced gating effects of Slc7a5 provides a more sensitive tool to probe the effects of Slc3a2. Based on BRET spectra (Fig. 6c) and variable gating properties of split-YFP tagged Kv1.2:Slc7a5:Slc3a2 complexes (Fig. 7), we had speculated that attenuation of the gating effects of Slc7a5 may

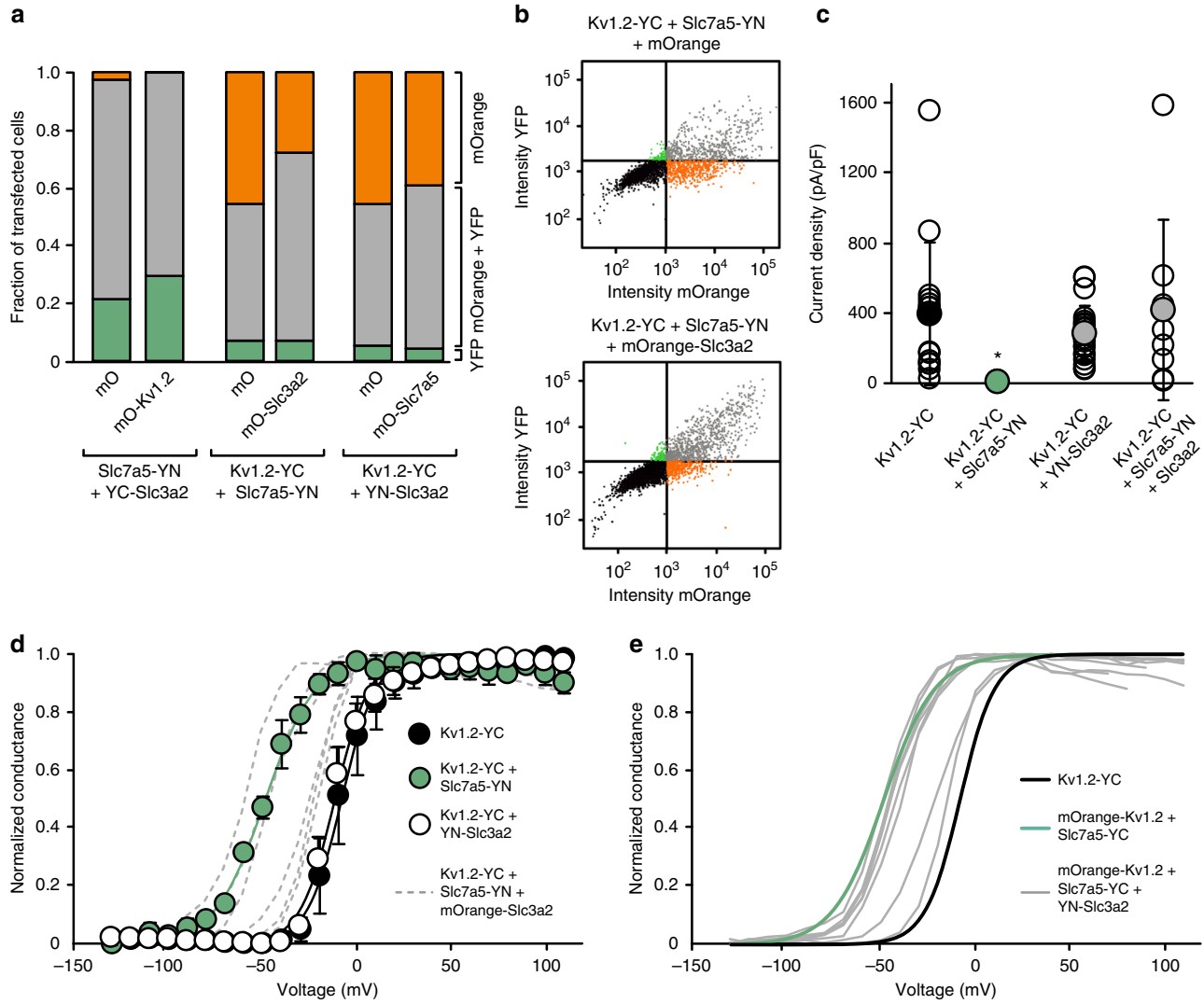

**Fig. 7** Using split YFP constructs to measure competition for Slc7a5 between Kv1.2 and Slc3a2. YFP fragments (YFPN residues 1–157, YN; and YFPC residues 158–239, YC) were fused to Kv1.2, Slc3a2, and Slc7a5 as indicated. **a** Fluorescence-activated cell sorting was performed on split-YFP tagged versions of Kv1.2, Slc7a5, and Slc3a2, with the third protein tagged with LSS-mOrange. All DNA transfections were done at a 1:1:1 ratio. Bars represent the distribution of transfected cells with prominent mOrange fluorescence (upper left quadrant in **b**), prominent YFP (lower right quadrant), or prominent signals from both fluorophores (upper right quadrant). **b** Representative data from FACS sorting of cells in the Kv1.2-YFPC + Slc7a5-YFPN + mOrange or +mOrange-Slc3a2 conditions. **c** Kv1.2-YFPC was expressed with combinations of Slc7a5-YFPN and Slc3a2 as indicated. The data from individual cells are plotted along with the mean ± s.d., $n = 7$–20. **d** Conductance-voltage relationships were generated for each condition after current disinhibition with a 30 s hyperpolarization to $-120$ mV (Kv1.2-YFPC $V_{1/2} = -8.4 \pm 7$, $n = 10$; Kv1.2-YFPC + Slc7a5-YFPN $V_{1/2} = -49 \pm 2$, $n = 5$; Kv1.2-YFPC + YFPN-Slc3a2 $V_{1/2} = -11 \pm 4$, $n = 11$). Individual conductance-voltage relationships for six cells transfected with Kv1.2-YFPC + Slc7a5-YFPN + Slc3a2 are depicted in gray (overall $V_{1/2} = -36 \pm 16$ mV). Data at each voltage are presented as mean ± s.d., $n = 4$–11. **e** Individual conductance-voltage relationships for seven cells transfected with mOrange-Kv1.2 + Slc7a5-YFPC + YFPN-Slc3a2 (transfection ratio 2:1:2)

not require direct physical occlusion/competition of the Kv1.2:Slc7a5 interaction by Slc3a2. In the case of the Kv1.2 R297Q or L298F mutants, it is clear that co-expression with Slc3a2 leads to incomplete rescue of gating properties, and wide variability of gating (Supplementary Figure 5a, c). This wide variation is likely due to the vast dynamic range of gating arising from the Slc7a5-mediated effects, together with cell-to-cell variability of relative expression of Slc7a5 and Slc3a2. In addition, we observed that co-expression with Slc3a2 led to incomplete rescue of channel deactivation (Supplementary Figure 5b, d). These findings support the idea that Slc3a2 and Slc7a5 can co-assemble and influence the gating properties Kv1.2.

## Discussion

In this study, we demonstrate profound regulation of Kv1.2 gating and expression by Slc7a5, in addition to its canonical function as an amino acid transporter[41,42]. Slc7a5 suppresses Kv1.2 by diminishing expression (Fig. 2), and a combination of gating effects that cause channels to accumulate in a non-conducting state that requires much hyperpolarized voltages for recovery to occur (Figs. 3–5). Recently reported Slc7a5 mutations linked to autosomal recessive inherited neurodevelopmental delay/autism spectrum disorder (A246V and P375L) have attenuated effects on Kv1.2 gating and current density, possibly due to a localization defect that prevents association with Kv1.2 (Fig. 8). In contrast, epileptic encephalopathy-linked mutations of

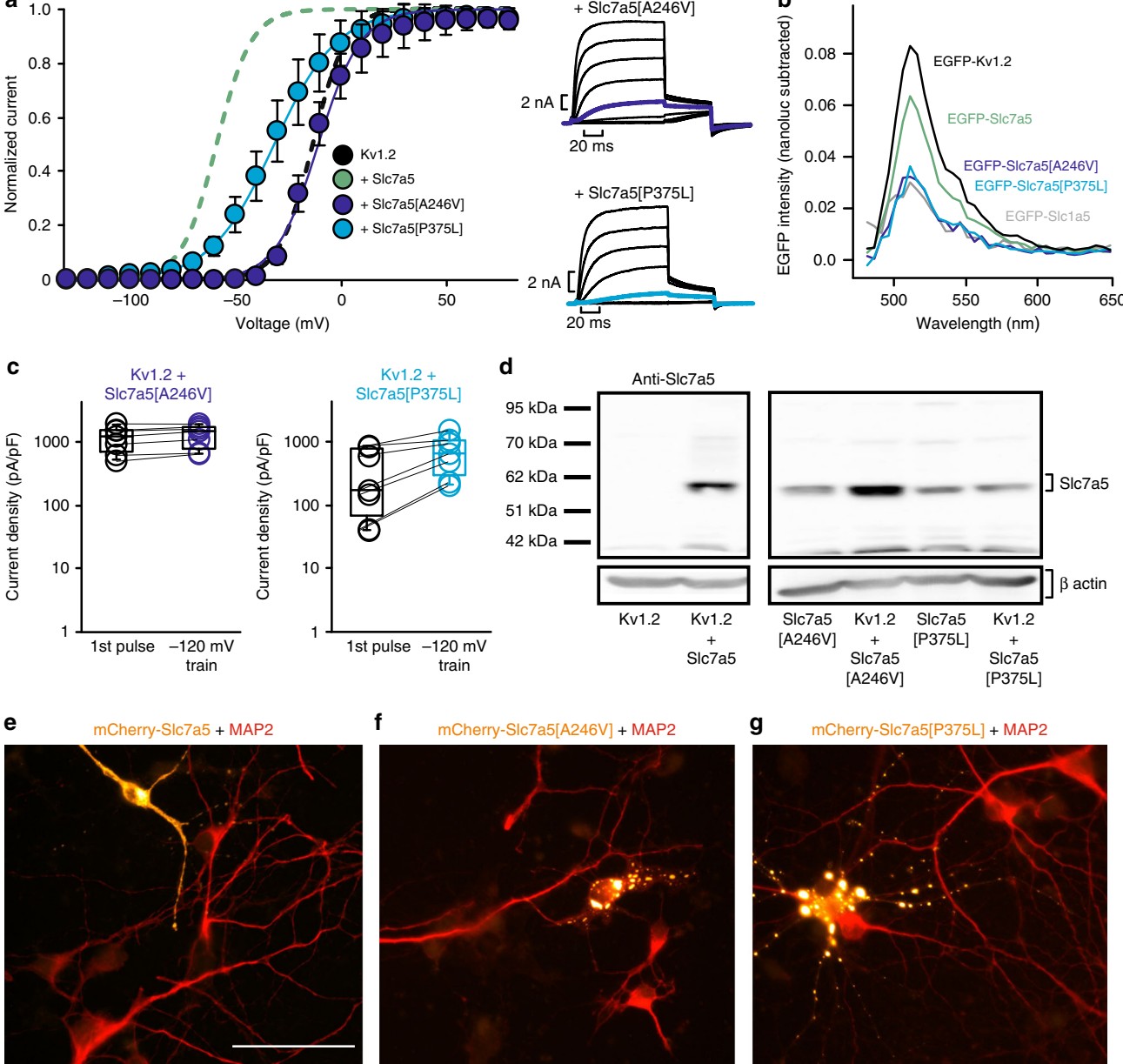

**Fig. 8** Disease-linked Slc7a5 mutations have attenuated effects on Kv1.2. **a** Kv1.2 channels were co-expressed with Slc7a5[A246V] or Slc7a5[P375L] (1:1) in *ltk*-mouse fibroblast cells. Conductance-voltage relationships were gathered as described in Fig. 1 (+Slc7a5[A246V] $V_{1/2} = -10 \pm 6$ mV, $k = 11 \pm 3$ mV, $n = 6$; +Slc7a5[P375L] $V_{1/2} = -32 \pm 8$ mV, $k = 16 \pm 8$ mV, $n = 10$). Dashed lines illustrate conductance-voltage relationships for WT Kv1.2 (black) and Slc7a5 (green) as a reference. **b** Representative BRET signals were calculated for mEGFP-tagged Slc7a5 mutants co-expressed with Kv1.2-nanoluc, as described in Fig. 6 (representative data from three experiments). **c** Current density at +60 mV was measured from Kv1.2 co-expressed with various Slc7a5 mutants, before and after a 30 s train of −120 mV hyperpolarizations to −120 mV. **d** Cell lysates were probed for Slc7a5 expression using western blot for cells transfected with Kv1.2 plus each of the Slc7a5 mutations. No significant changes were detected for expression of the Slc7a5 mutants ($n = 3$, $p > 0.05$). **e–g** Cortical and hippocampal neurons from P2 rat pups transiently transfected at 7 days in vitro with either mCherry-Slc7a5 WT or mCherry-Slc7a5 [A246V] or [P375L], and at 10 days in vitro, fixed with 4% paraformaldehyde, permeabilized with 1% Triton X-100 and stained with anti-MAP2 primary antibodies and fluorescent secondary antibodies. Images are representative of three neuronal cultures. Scale bars represent 50 μM

Kv1.2 (R297Q and L298F) enhance susceptibility to the gating effects of Slc7a5 (Fig. 9). Relative to other known accessory subunits of Kv1 channels, these gating effects are unique and large in magnitude, and reveal a previously unrecognized ion channel regulatory mechanism[12,15,16,43].

Kv1.2 was the first eukaryotic voltage-gated ion channel with a reported atomic resolution structure, and has served as a valuable template for understanding mechanisms of voltage-dependent

gating of ion channels[4]. In search of additional regulatory mechanisms, we performed mass spectrometry of cross-linked Kv1.2 complexes, followed by screening to identify proteins with clear effects. We used cross-linking to maximize detection of proteins that may have transient or lower affinity interactions with the channel, or that are lost with detergent solubilization. Our approach was less stringent at the immunoprecipitation stage, at the expense of more laborious screening of candidate

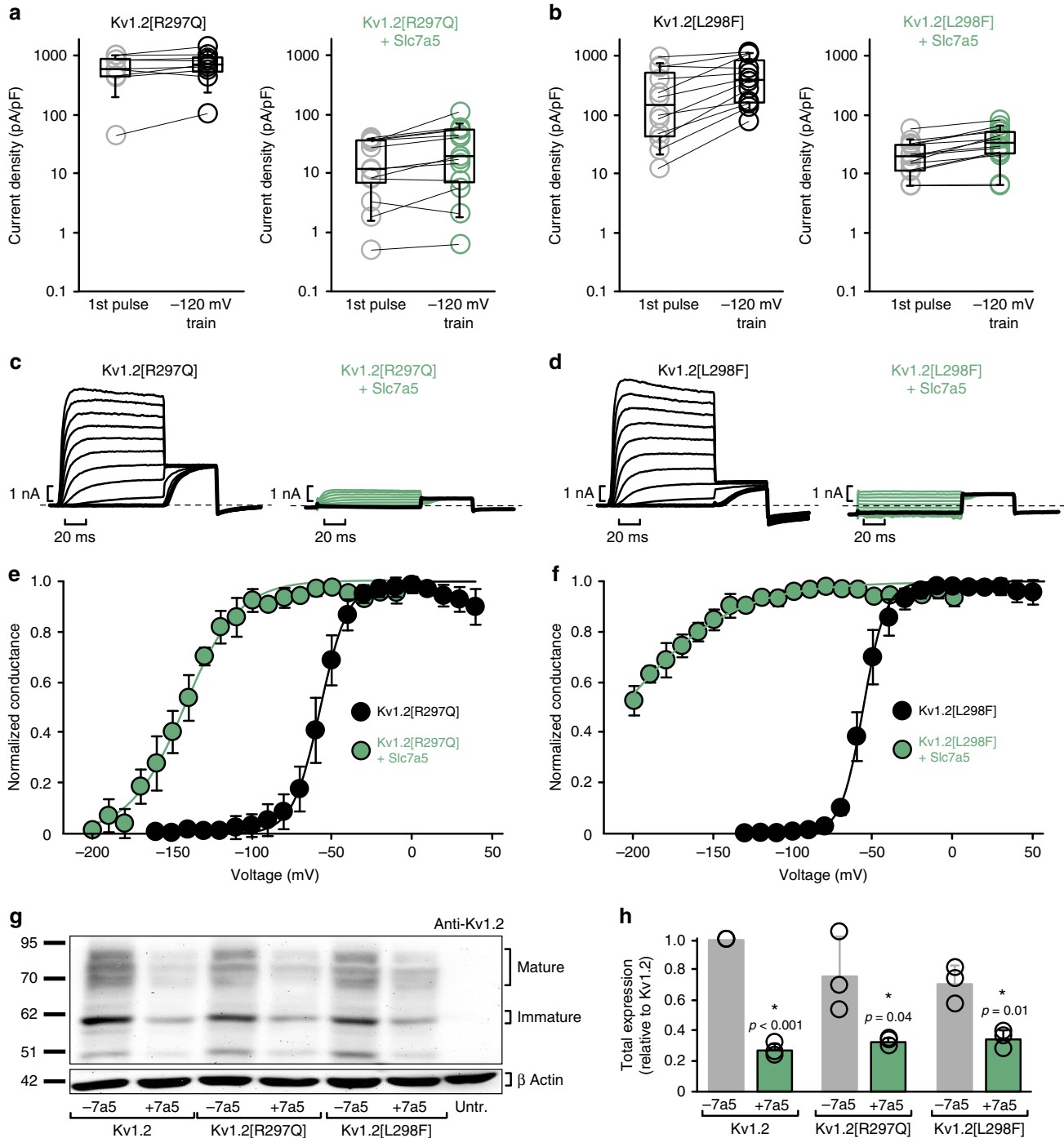

**Fig. 9** Kv1.2 disease-linked mutations are strongly suppressed by Slc7a5. **a**, **b** Current density at +60 mV was measured before and after a 30 s hyperpolarizing train to −120 mV for Kv1.2[R297Q] alone ($n = 6$) or with Slc7a5 ($n = 13$), and Kv1.2[L298F] alone ($n = 12$) or with Slc7a5 ($n = 14$), as indicated. **c**, **d** Exemplar currents recorded from Kv1.2[R297Q] or Kv1.2[L298F], expressed in *ltk-* mouse fibroblasts. Highlighted sweeps are to −120 mV with a tail current voltage of −30 mV. **e**, **f** Conductance-voltage relationships for Kv1.2[R297Q] ($V_{1/2} = −57 \pm 4$ mV, $k = 8 \pm 1$ mV, $n = 6$) and Kv1.2[R297Q] + Slc7a5 (2:1 transfection, $V_{1/2} = −143 \pm 6$ mV, $k = 17 \pm 2$ mV, $n = 9$), Kv1.2[L298F] ($V_{1/2} = −56 \pm 1$ mV, $7 \pm 1$ mV, $n = 4$) and Kv1.2[L298F] + Slc7a5 (4:1 transfection, $V_{1/2} < −200$ mV, estimated to be −210 ± 5 mV, $k = 35 \pm 6$ mV, $n = 9$). **g** Exemplar western blot of cell lysates from *ltk-* mouse fibroblasts transfected with indicated combinations of Kv1.2 mutants and Slc7a5 as indicated, each at a 1:1 transfection ratio. **h** Densitometric quantification of Slc7a5 effects on total protein levels of various Kv1.2 mutants (data shown as mean ± s.d., $n = 3$ per condition); Student's *t*-test was used to compare expression of each mutant in the presence and absence of Slc7a5

genes. A more stringent protocol (non-cross-linked, more detergent washes, etc.) would likely yield less candidate genes, but might also miss important transient or low affinity interactions.

The Slc7a5-mediated hyperpolarizing shift of channel activation, coupled with suppression of current, initially seemed to be counteracting effects. However, these are related—the hyperpolarizing shift contributes to current suppression by making the channels more prone to enter an inactivated (or other non-canonical non-conducting) state. This effect of Slc7a5 is accomplished by the combination of opening at more negative voltages

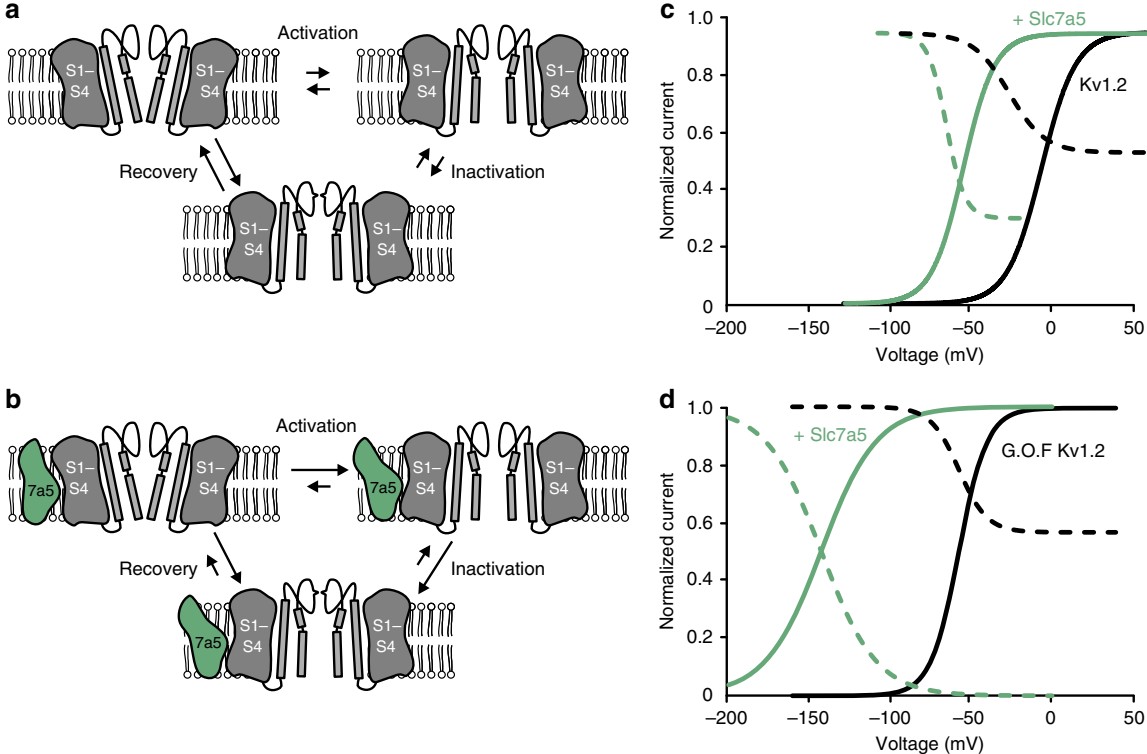

**Fig. 10** Inactivation trap gating effect of Slc7a5 on Kv1.2. **a** Generic gating scheme of Kv1.2 cycling between closed, activated, and inactivated states. **b** Slc7a5 accelerates channel activation and inactivation, and slows recovery from inactivation, leading to accumulation of channels in an inactivated state. **c** Activation curves from Fig. 4 and inactivation curves from Fig. 5 are shown, illustrating the greater propensity for inactivation of Kv1.2 when assembled with Slc7a5. **d** Disease-linked gain-of-function mutants (GoF Kv1.2) exhibit greatly exaggerated sensitivity to Slc7a5 gating effects, such that Kv1.2 channels in complex with Slc7a5 are predominantly inactivated at resting voltages. Activation curves are reproduced from Fig. 9e. Inactivation curves are hypothesized, as they could not be measured due to the extreme voltages required

(stabilization of channel activation) and enhancing the inactivation rate (accentuated in the V381T mutant, Fig. 10a, b). In the presence of Slc7a5, channel activation and inactivation are sufficiently shifted (Fig. 10c, green lines) that some inactivation (or other non-conducting mode) of Kv1.2 may occur even when cells are at rest. In addition, extremely negative voltages are required to allow for current recovery/disinhibition (Fig. 3a, holding potential of −120 mV). In a physiological system where such extreme voltages are never reached, this molecular complex would act as a trap that could alter excitability by silencing channels. Although the stoichiometry of the Kv1.2:Slc7a5 interaction is not yet known, it is important to note that assembly with different numbers of Slc7a5 subunits, or heteromeric channels with Slc7a5 sensitive (e.g. Kv1.2) and insensitive (e.g. Kv1.5) subunits, may lead to intermediate instances of the effects described. Our findings likely represent an extreme case, and variable or regulated assembly might temper these effects in native settings. Further exploration of the entire Kv1 family, other Kv subtypes, and the Slc7 family will likely reveal functional diversity of these effects. In addition, a deeper investigation of the variable influence of Slc3a2 may provide important additional understanding of this ion channel regulatory mechanism.

The silencing effect of Slc7a5 may be noteworthy in the context of an apparent paradox in the characterization of epilepsy-associated Kv1.2 mutations. Recent findings have reported similar epileptic phenotypes for both gain- and loss-of-function Kv1.2 mutants[9,10]. In addition, an observation lacking a good explanation has been that Kv1.2 mutants with powerful gain-of-function (increased current magnitude and ~50 mV hyperpolarizing gating shifts) lead to hyperexcitability, rather than

suppression of excitation[44]. Although many factors may contribute to these outcomes, gain-of-function mutations of Kv1.2 are much more sensitive to the Slc7a5-mediated silencing, and this could lead to a loss-of-function effect when co-expressed with Slc7a5 (even in the presence of Slc3a2, Supplementary Figure 5). The hypersensitivity of Kv1.2 gain-of-function mutations arises from two main differences that we have observed. First, there is an intrinsic gating shift of the mutant channels (Fig. 10d, black lines), and for reasons that are not yet apparent, the Slc7a5-mediated gating shift is even more extreme than seen with WT Kv1.2 (Fig. 10d, green lines). Therefore, a greater fraction of Kv1.2 channels would activate and become silenced/inactivated at physiological resting voltages. Second, this extreme gating shift induced by Slc7a5 causes the gain-of-function mutants to require extreme hyperpolarized voltages for disinhibition. The unique gating effects of Slc7a5 provide an interesting example of how the outcome of a disease-linked ion channel mutant might be fundamentally altered by assembly with an accessory protein.

Slc7a5 is a multi-pass transmembrane protein[36], a feature that stands out from previously described accessory proteins of Kv1 channels, which are predominantly soluble cytoplasmic proteins, and often associated with the cytoskeleton[13,15,45]. A prominent area of investigation of Slc7a5 has been its expression in vasculature where it transports amino acids, neurotransmitters and small drugs across the blood–brain barrier[31,37,46]. While it is enriched in the epithelial cells of blood vessels in the brain, it is also expressed midbrain structures, cortex and hippocampus where its function is less well characterized[47]. Pathological consequences of Slc7a5 disruption (mutation or knockout) are predominantly attributed to its amino acid transport function. Our

findings indicate that future consideration of a role of Slc7a5 in ion channel regulation is also warranted, in terms of physiological consequences of the Slc7a5:Kv1.2 interaction. However, further investigation of this complex will be required to generate tools that distinguish the amino acid transport function vs. ion channel regulation. It will also be of interest to determine whether Kv1.2 exerts a reciprocal effect on Slc7a5 transport activity. The assembly of ion channels with membrane proteins that have pleiotropic functions is an emerging theme, as other transporter subunits have recently been reported to influence ion channels. One example is a functional interaction of the sodium chloride cotransporter (NCC) pump and the endothelial sodium channel (ENaC) at the distal convoluted tubule[48]. Also, the sodium-coupled myoinositol transporter SMIT1 affects expression of KCNQ1 and KCNQ2/3 potassium channels[49,50]. These findings highlight the importance of continuing to investigate interactions of Kv channels with accessory subunits, and how these shape potassium currents in excitable cells and may influence the functional outcome of disease-linked mutations.

## Methods

**Co-immunoprecipitation**. We generated a bait construct (Kv1.2-1D4) in pcDNA3.1(-) comprising the N-terminus of Kv1.5 (residues 1–121), and the transmembrane domains and C-terminus of Kv1.2[S371T] to increase cell surface and overall expression[51]. This hybrid channel construct also included a C-terminal 1D4 epitope tag[52]. Ten centimeter of dishes of HEK cells were transfected with Kv1.2-1D4 and harvested after 72 h of growth in a 37 °C 5% $CO_2$ incubator. Cells were then either incubated in 1 mM DTT or ambient redox PBS (10 mM $PO_4^{3-}$, 137 mM NaCl, and 2.7 mM KCl, pH 7.4) for 1 h. To harvest protein, cells were washed with PBS, then incubated for 30 min in 250 μM ethylene glycol bis(succinimidyl succinate), a 16 Å long bifunctional cross-linker to link nearby free amines, diluted in PBS. Cells were lysed with 20 mM HEPES, 0.1 M NaCl, 2 mM $MgCl_2$, 20 mM CHAPS, and protease inhibitor at 4 °C for 20 min. Lysates were centrifuged at 21,130×g for 5 min.

1D4 antibody-coated beads were prepared for affinity purification by incubating CNBr activated sepharose 4B beads (GE Healthcare) with 1D4 monoclonal antibody (Rho-1D4 purified monoclonal antibody, Flintbox) according to the manufacturer's instructions. Sixty microliters of beads was washed with 500 μL of wash buffer (20 mM HEPES, 0.1 M NaCl, 2 mM $MgCl_2$, 10 mM CHAPS). Supernatant from cell lysates was added to the beads and incubated at 4 °C for 20 min. Beads were then washed with wash buffer six times. Proteins were then eluted with 60 μL of elution buffer (wash buffer +2 mg/mL 1D4 peptide) at 10 °C for 10 min. All chemicals were purchased from Sigma-Aldrich or Fisher.

**Mass spectrometry**. Cross-linked protein samples were analyzed by mass spectrometry at the Proteomics Core Facility at the University of British Columbia (Vancouver, Canada). Protein precipitates were boiled in SDS sample buffer and run on a short 10% (wt/vol) SDS/PAGE gel. Proteins were visualized by colloidal coomassie and in-gel digested. Post-tryptic peptides from three different conditions (Kv1.2-1D4, Kv1.2-1D4 +1 mM DTT, untransfected) were labeled with stable isotopic dimethyl labels for quantitation of relative peptide abundance in each sample[53]. Samples were mixed and analyzed by a quadrupole–time of flight mass spectrometer (Impact II; Bruker Daltonics) coupled to an Easy nano LC 1000 HPLC (Thermo Fisher Scientific)[54]. Analysis of mass spectrometry data was performed using MaxQuant 1.5.3.30. The search was performed against a database comprised of the protein sequences from Uniprot *Homo sapiens* sequence entries plus common contaminants with variable modifications of methionine oxidation, and N-acetylation of the proteins, in addition to the isotopes of dimethyl modifications for quantitation. Only those peptides exceeding the individually calculated 99% confidence limit (as opposed to the average limit for the whole experiment) were considered as accurately identified[54]. The mass spectrometry proteomics data have been deposited to the ProteomeXchange Consortium via the PRIDE partner repository with the dataset identifier PXD011010[55]. The amount of each protein was quantified relative to the others based on the relative abundance of each protein tagged with the three different dimethyl tags. Candidate interacting proteins were selected and prioritized manually based on abundance relative to untransfected control samples, and screening against the CRAPome[27]. cDNAs for candidate interactors were purchased from the DNASU plasmid repository, and subcloned into pEGFP-C1 using PCR to introduce compatible restriction sites.

**Cell culture and expression**. cDNAs were expressed using the pcDNA3.1(-) vector (Invitrogen), mEGFP-C1, a gift from Michael Davidson (Addgene plasmid #54759) or pLSS-mOrange-C1, a gift from Vladislav Verkhusha (Addgene plasmid #37131)[56]. Where indicated, fluorescent proteins were fused Slc7a5 and Slc3a2 using standard PCR and compatible restriction digestion and ligation. Constructs

were all verified by diagnostic restriction digestions and Sanger sequencing (Genewiz, Inc.).

Mouse *ltk-* fibroblast cells (ATCC) were used for patch clamp experiments and western blots, except in Fig. 4c–e, where HEK293 cells (ATCC) were used. Cells were maintained in culture in a 5% $CO_2$ incubator at 37 °C in DMEM supplemented with 10% FBS and 1% penicillin/streptomycin. Cells were split onto sterile glass coverslips and transfected with cDNA using jetPRIME transfection reagent (Polyplus). Fluorescent proteins were used to identify transfected cells for electrophysiological recording. Recordings were done 24–48 h following transfection.

**Mutagenesis**. Kv1.2 and Slc7a5 mutagenesis was done by overlap extension PCR with two mutagenesis primers, one in the 5′ direction in addition to a standard SP6 3′ flanking primer, and another mutagenesis primer in the 3′ direction in addition to a standard T7 5′ flanking primer[57]. Primers for each construct are listed in Supplementary Table 1. All constructs were expressed using the pcDNA3.1(-) vector (Invitrogen), and verified by diagnostic restriction digestions and Sanger sequencing (Genewiz, Inc. or University of Alberta Applied Genomics Core).

**Electrophysiology**. Patch pipettes were manufactured from soda lime capillary glass (Fisher), using a Sutter P-97 puller (Sutter Instrument). When filled with standard recording solutions, pipettes had a tip resistance of 1–3 MΩ. Recordings were filtered at 5 kHz, sampled at 10 kHz, with manual capacitance compensation and series resistance compensation between 70 and 90%, and stored directly on a computer hard drive using Clampex 10 software (Molecular Devices). Bath solution had the following composition: 135 mM NaCl, 5 mM KCl, 1 mM $CaCl_2$, 1 mM $MgCl_2$, 10 mM HEPES, and was adjusted to pH 7.3 with NaOH. Pipette solution had the following composition: 135 mM KCl, 5 mM K-EGTA, 10 mM HEPES and was adjusted to pH 7.2 using KOH. Chemicals were purchased from Sigma-Aldrich or Fisher.

**Electrophysiology data analysis**. Throughout the text we have displayed data for all individual cells collected, in addition to reporting mean ± SD or a box plot with the median ± 95% CI. Conductance-voltage relationships were fit with a Boltzmann equation (Equation 1), where $I/I_{max}$ is the normalized current, $V$ is the voltage applied, $V_{1/2}$ is the half-activation voltage, and $k$ is a fitted value reflecting the steepness of the curve.

$$\frac{I}{I_{max}} = \frac{1}{1 + e^{-(V-V_{1/2})/k}}. \tag{1}$$

Conductance-voltage relationships were fit for each individual cell, and the extracted fit parameters were used for statistical calculations. Where shown, box plots depict the median, 25th and 75th percentile (box), and 10th and 90th percentile (whiskers).

**Bioluminescence resonance energy transfer**. Nanoluc was amplified from pcDNA3.1-ccdB-Nanoluc (gift from Mikko Taipale, Addgene plasmid # 87067), and fused to the Kv1.2 C-terminus in pcDNA3.1(-) using EcoRI and HindIII restriction sites. Other cDNAs were tagged at the N-terminus with mEGFP using standard subcloning methods. HEK cells were transiently transfected with cDNAs encoding BRET donors and acceptors for 48 h, then replated onto white polystyrene 96-well plates (Thermo Fisher). After 24 h, cells were washed with PBS, and incubated with Nano-Glo live cell assay reagent (Promega). Emission spectra were measured between 400 and 700 nm in 2 or 5 nm increments, for 2 s at each interval, with a Synergy H4 Hybrid Reader (BioTek). Spectra were normalized to the peak nanoluc emission, and the normalized Kv1.2-nanoluc spectrum (measured in parallel) was subtracted to obtain the mEGFP emission. Integrated mEGFP emission (area under the curve) was normalized to the integrated mEGFP emission from Kv1.2-nanoluc + mEGFP-Kv1.2 in each experiment.

**Western blot**. Cell lysates from transfected *ltk-* fibroblasts were collected in NP-40 lysis buffer (1% NP-40, 150 mM NaCl, 50 mM Tris-HCl) 3 days after transfection, separated using 8% SDS-PAGE gels, and transferred to nitrocellulose membranes using standard methods. Kv1.2 was detected using a mouse monoclonal Kv1.2 antibody (1:10,000 dilution, clone #K14/16 75–008; NeuroMab) and HRP-conjugated goat anti-mouse antibody (1:30,000 dilution, SH023; Applied Biological Materials). Slc7a5 was detected using a rabbit polyclonal Slc7a5 antibody (1:500 dilution, KE026; Trans Genic Inc.) and HRP-conjugated goat anti-rabbit antibody (1:15,000 dilution, SH012; Applied Biological Materials). Chemiluminescence was detected using SuperSignal West Femto Max Sensitivity Substrate (Thermo Fisher Scientific) and a FluorChem SP gel imager (Alpha Innotech). Uncropped western blot images used for figure preparation are included in Supplementary Figures 6–9.

**RNA isolation and RT-PCR**. Total RNA was extracted from enriched rat cortical neuron cultures using TRIzol reagent (Life Techologies), and reverse transcription was performed using the RETROscript reverse transcription kit (Ambion). cDNAs

of interest were amplified using Phusion polymerase (Thermo Fisher). Slc7a5 cDNA was detected using the primers acctgcaccagaagttgtcc (forward) and gtgaagtaggccaggttcg (reverse). Slc3a2 cDNA was detected using the primers ccaagttcgggatgtgggaa (forward) and tgcatgctccccagtgaaaa (reverse).

**Fluorescence-activated flow cytometry.** HEK cells were transfected with the indicated constructs for 72 h. Cells were washed with PBS, trypsinized for 5–10 min, resuspended in DMEM then spun down, resuspended in PBS and spun down and finally resuspended in PBS + 2% FBS + 2 μM EDTA. Samples were measured using an Attune NxT Flow Cytometer (Flow Cytometry Core Facility, University of Alberta, Edmonton, Canada) and analyzed using the FlowJo program.

**Neuron isolation and fixation.** Hippocampi or cortices from postnatal day 0–2 Sprague Dawley rats were isolated and digested in 27 U/mL papain solution in HEPES-buffered HBSS (10 mM HEPES, 1 mM sodium pyruvate, 1% penicillin/streptomycin dissolved in calcium/magnesium-free HBSS) for 10 min in a 37 °C water bath. DNase (0.15 mg/mL) was added to the digestion mixture and incubated for 5 min in the 37 °C water bath. Ten percent FBS was then added and the mixture was centrifuged at 200×g for 1 min. The supernatant was aspirated and the pellet was resuspended in plating media (10% FBS, 1 mM sodium pyruvate, 2 mM glutamine and 1% penicillin/streptomycin in MEM) and triturated. The mixture was filtered with a 40 μM sterile cell strainer. The neurons were counted using a hemocytometer and plated at a density of 800 cells/mm². After 3 h, the plating media were replaced with growth media (1X B27 supplement, 500 μM GlutaMAX-1 and 1% penicillin/streptomycin in Neurobasal-A). Neurons were treated with cytosine arabinoside (5 μM) for 24 h at 3 days in vitro. All reagents and materials were purchased from Fisher, Sigma-Aldrich and Invitrogen.

Immunohistochemistry experiments were done at 7–9 d in vitro. Neurons were fixed with 4% paraformaldehyde, permeabilized with 1% Triton X-100 and stained with primary antibodies as follows: mouse monoclonal Kv1.2 antibody (1:500 dilution, clone #K14/16 75–008; NeuroMab), rabbit polyclonal Slc7a5 antibody (1:50, KE026; Trans Genic Inc.), mouse or rabbit NF-200 antibody (1:500 dilution, N4142 or N5389, Sigma-Aldrich). Neurons were then stained with secondary antibodies as follows: anti-rabbit Alexa 680 (cat. A21109, Invitrogen), anti-mouse Alexa 488 (cat. A21202, Invitrogen), and/or anti-mouse Alexa 555 (cat. A31570, Invitrogen) each at a 1:500 dilution. Neuronal images were obtained on a Zeiss Colibri microscope and analyzed using ImageJ software. All procedures on animals were approved by the University of Alberta's Animal Care and Use Committee and were in accordance with the guidelines of the Canadian Council on Animal Care.

## Data availability

Data supporting the findings of this manuscript are available from the corresponding author upon reasonable request. The mass spectrometry proteomics data have been deposited to the ProteomeXchange Consortium via the PRIDE partner repository with the dataset identifier PXD011010. Plasmids and other non-commercial reagents are available upon request.

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

## Acknowledgements

This work was supported by a Canadian Institutes of Health Research project grant (CIHR PS 148815) and Human Frontiers in Science Program Young Investigator Award (RGY-0081) to HTK. V.A.B. was supported by a Vanier Canada Graduate Scholarship and the University of British Columbia MD/PhD program. H.T.K. was supported by a Canadian Institutes of Health New Investigator salary award, and the Alberta Diabetes Institute. We are grateful for the assistance of the University of British Columbia Proteomics Core Facility.

## Author contributions

V.A.B., R.Y.Y., and H.T.K. conceived of the study. V.A.B., R.Y.Y., and L.C.M. performed experiments. S.S. and L.C.M. contributed critical reagents and expertise. V.A.B. and H.T. K. analyzed data and wrote the initial draft of the manuscript. All authors assisted with revision and approval of the final manuscript.

## Additional information

**Competing interests:** The authors declare no competing interests.

