## [Peer Review File · Nature Communications]

Reviewers' Comments:

Reviewer #1:

Remarks to the Author:

In this manuscript, Baronas and colleagues report the results of co-immunoprecipitation (following cross-linking), coupled with mass spectrometry, designed to identify potential "binding partners" of Kv1.2, followed by functional screening of the effects of the identified interacting proteins on the expression and biophysical properties of Kv1.2-encoded channels.

They report the identification of multiple Kv1.2-interacting proteins, one of which they went on to study in detail. This protein was Slc7a5, a neutral amino acid transporter. Experiments are presented demonstrating that co-expression of Slc7a5 with Kv1.2 reduces total Kv1.2 protein, and dramatically hyperpolarizes the voltage-dependence of activation of Kv1.2-encoded currents (by > 40 mV). They also show that the effects of Slc7a5 are attenuated with the additional co-expression of Slc3a2, a previously identified binding partner of Slc7a5. These latter observations were interpreted as reflecting Slc3a2-mediated reduced binding of Slc7a5 to Kv1.2, although this was not demonstrated directly. The reduction in Kv1.2-encoded currents with co-expression of Slc7a5 was attributed to an interesting and seemingly novel 'inactivation trap' mechanism.

In the final series of experiments, the authors go on to examine the effects of two recently reported variants of Slc7a5, linked to neurodevelopmental delay, and epilepsy-linked gain-of-function Kv1.2 mutants on Slc7a5-mediated modulation of heterologously expressed wild type and mutant Kv1.2-encoded currents. The work is carefully done and well-described. The data are of high quality and the manuscript is well-written.

The authors might well be correct that the interaction between Slc7a5 and Kv1.2 and the observed functional effects of Kv1.2 currents "may influence neuronal function and disease pathogenesis in patients with Slc7a5 or Kv1.2 mutations". Much more would need to be done, however, to demonstrate the physiological and/or the pathophysiological relevance/significance of the observations reported here. In the absence of experiments documenting neuronal expression of Slc7a5 and the functional effects of Slc7a5 co-expression on Kv1.2-homotetrameric or Kv1.2-heterotetrameric channels in neurons, the study represents little more than a well done heterologous co-(over)expression study in unknown physiological relevance/significance.

Reviewer #2:

Remarks to the Author:

Summary:

Baronas and colleagues report that a previously unknown protein interaction between the Kv1.2 potassium channel and a neutral amino acid transporter (Slc7a5 / LAT-1) causes a marked suppression of current through a novel mechanism. The Kv1.2 / Slc7a5 interaction enhances C-type inactivation (faster, more complete) and produces an enormous hyperpolarizing shift (-47 mV) of activation such that, in combination, these effects result in an "inactivation trap" over a tested voltage range of -100 mV to -80 mV. It should be noted that the hyperpolarizing shift of activation alone is quite remarkable. I am not aware of any other channel subunit or cellular modification (e.g. phosphorylation, redox, lipid interaction) that is capable of causing such a large shift. The protein interaction is specific (e.g. current modification does not occur for Kv1.5; nor with Slc1a5 or Slc3a2 co-expression although the latter partially mitigates the Slc7a5 effect). Proximity of the Kv1.2 / Slc7a5 interaction is demonstrated with bioluminescence resonance energy transfer (BRET), and a "split" YFP reporter is used to show that the mitigating effect of Slc3a2 is not caused by displacement of Slc7a5. The final section of the paper highlights the potential significance of this new finding to human disease. Kv1.2 missense mutations associated with epileptic encephalopathy were previously reported to cause gain-of-function changes, which is difficult to reconcile with the clinical phenotype since a reduction of neuronal K conductance would

be expected to be epileptogenic. The present study shows two different Kv1.2 mutant channels have exaggerated sensitivity to current suppression by Slc7a5, thereby implicating a possible loss-of-function pathogenesis. Conversely, missense mutations of Slc7a5 associated with recessively-inherited autism spectrum disorder greatly attenuated the suppression of Kv1.2 current or in one case completely abolished the suppression.

Critique:

The remarkable findings of this study will be of high interest to biophysicists interested in regulation of ion channel gating and have far-reaching implications for the characterization and interpretation of disease-associated channelopathies, when studied in artificial expression systems. A number of questions need to be resolved (by comment or additional experiments) to more fully understand the potential biological implication of this novel protein-protein interaction.

1) The L-type 1 neutral amino acid transporter is a dimer of LAT1/CD98 (Slc7a5 / Slc3a2 gene products) cross-linked by a sulfhydryl bond. The profound effects on Kv1.2 gating were observed for LAT1 expressed alone, and were markedly attenuated by co-expression of CD98. In light of this, (a) would the (unbound) LAT1 protein be present in sufficient quantities to produce the gating effects on Kv1.2 channels in cells normally expressing the transporter, and (b) exposure to reducing agents (DTT or DTE) should be tested to see if this eliminates the mitigation of suppression attributed to Slc3a2 (CD98) co-expression.

2) Related to item (1), the authors should test whether the profoundly enhanced suppression of current for the Kv1.2 mutations associated with epilepsy (Fig 9E, F) is prevented by co-transfection with Slc3a2. If co-expression of Slc3a2 largely prevents this loss-of-function effect, then the relevance to disease pathomechanism becomes uncertain.

3) LAT1 / Slc7a5 is clearly expressed "in the brain", but is it known whether expression occurs in neurons (with Kv1.2)? This is an important question since the highest expression in the brain is probably from endothelial cells to mediate amino acid transport across the blood brain barrier. If LAT1 is not in neurons, then the interesting findings in this study do not have much biological relevance (although they still identify an interesting biophysical effect on channels).

4) What is the potential role of the KvBeta subunit in these phenomena? Do ltk- cells express KvBeta subunits? Does co-transfection of KvBeta alter the gating effects produced by Slc7a5? This is an important question because if the KvBeta subunit prevents the Slc7a5 suppression of current, then the biological relevance is again in question.

Specific Points:

1) Statistical tests of the perceived differences in current density (Fig 2a) are not provided. The authors are commended for showing the high variance of current density that occurs with transient transfections, but this also highlights the need for ANOVA.

2) The effect of Slc7a5 co-transfection on Kv1.2 inactivation is a key contributor to the mechanism of the "inactivation trap" and understanding the relief of current suppression by hyperpolarization (Fig 3). As such, Supplementary Fig 2 showing the voltage dependence of inactivation should be moved to the main body of the paper. In addition, there may be a problem with the voltage protocol used to measure pre-pulse inactivation (Supplementary Fig 2). The inactivation curve for Slc7a5 (Fig S2) shows fully available channels (i.e. not inactivated) at about -80 mV, with a flat plateau to -110 mV. The data in Fig 3, however, show a large difference in available current produced by a holding potential of -120 mV versus -100 mV or -80 mV. This difference is not because of the Kv1.2[V381T] background used to promote C-type inactivation (Fig S2), because the data in Fig 5B (Kv1.2[V381T] channel) also show voltage-dependent relief of current suppression at -120 mV compared to -100 mV. So either the relief of suppression by

hyperpolarization is not attributable to recovery from inactivation or the voltage-dependence of inactivation curve in Fig S2 is in error.

3) The holding potential used for the data in Fig 2A should be stated, since the Slc7a45 suppression of Kv1.2 current is voltage-dependent.

Reviewer #3:

Remarks to the Author:

This paper reports two interesting new findings that are worthy of attention:

A. A functionally significant interaction between an amino acid transporter and a voltage-gated ion channel.

- the full mass spec data should be provided as supplementary material or deposited in a public database.

B. The ability of co-expression of Slc7a5 to turn apparent gain-of-function-producing mutations into an effective loss of function. This is an interesting possible explanation for the otherwise puzzling observation that these mutations produce similar phenotypes in animals or humans to loss-of-function mutations.

However there are other aspects of the manuscript that are problematic and should be clarified or improved:

1. Is the electrophysiology done with the same 1.5/1.2 chimera that was used for the immunoprecipitation? If so, the abstract and text should reflect this. Also, some critical experiments would need to be repeated to support the case that native Kv1.2 participates in this same interaction.

2. The use of the expression "inactivation trap mechanism" is misleading to channel biophysicists: it implies a mechanism involving retention of the Kv1.2 channels in the inactivated state. The association of Slc7a5 does not appear to function in this way; instead it promotes inactivation by favoring entry into the inactivated state, both by favoring entry into the open state and by speeding up open state inactivation. It would be better simply to say that it promotes inactivation. It might be even better to title the manuscript based on point #2 above.

3. The authors overstate several aspects of the story involving Slc3a2 rescue of the Slc7a5 phenotype.

- there is no basis for talking about the relative strength of the Kv1.2-Slc7a5 interaction relative to the Slc3a2-Slc7a5. There is no information about the amounts of proteins, only the amount of plasmid DNA.

- there is no indication that Slc3a2 co-expression produces a statistically significant reduction in Kv1.2 BRET with Slc7a5 (Fig. 6C; other pairs are marked with p-values but not that one; the data points shown do not support a p-value < 0.05). If not, there is no place for a statement that it does.

4. The split YFP experiments do not necessarily suggest "that Slc3a2 does not prevent assembly of Slc7a5 and Kv1.2." If the membrane proteins associate more reversibly than the split YFP (and other results imply that this is so), the untagged Slc3a2 should lose out to the tagged Slc7a5, irrespective of their behavior without the split YFP.

5. Mutants of Slc7a5 may affect folding or expression of the transporter protein. Without controls to test this, no conclusions should be made about the reduced effect on Kv1.2. It is in any case a stretch to imply that the disease consequences of these mutations to the changes in Kv1.2 rather

than to the primary effect on amino acid transport.

6. What is the authors' hypothesis to explain how Slc7a5 coexpression reduces the level of Kv1.2 protein (especially given that there is no change in the relative amount of surface expression)?

7. Short of doing experiments in actual neurons, the authors could make a better case that these two proteins (Kv1.2 and Slc7a5) are likely to be co-expressed and associated in neurons. Transcriptomics data (Barres and Linnarsson publications) could help.

8. The statement that attention to Kv1.2 as a source of structural information may have interfered with studies of its physiology has no basis and serves no purpose. It should be removed.

Thank you to all reviewers for their careful reading and thoughtful comments. We have addressed all comments in a point-by-point manner below, along with multiple revisions and additions to the manuscript. We have tried to be as comprehensive as possible with our revisions, although we recognize that there will be much more work to characterize the physiology and underlying mechanisms of these effects.

A common thread related to the nature of the mutual effects of Slc3a2, Slc7a5, and Kv1.2. We wanted to highlight here that the experimental conditions (for the Slc3a2 gating rescue effect) were chosen to highlight that dramatic variability of channel gating can be achieved by this set of proteins. This is not a ‘black or white’ effect, and will require much more work to describe and understand in detail. The relative expression of these proteins can significantly affect the experimental outcome, as in our exploration of these effects we have found that greater expression of Slc7a5 leads to greater suppression of Kv1.2, and hinders the rescue effect of Slc3a2. We suspect that other factors, such as the relative expression of other Slc7 subtype, might also influence the effectiveness of Slc3a2 to rescue the gating effects on Kv1.2. Overall, there are clearly many more avenues to pursue related to this topic, but we have tried to be as comprehensive as possible with this original report, and hopefully generate new directions for investigation.

Thank you again for your time and valued input.

Reviewer #1 (Remarks to the Author):

In this manuscript, Baronas and colleagues report the results of co-immunoprecipitation (following cross-linking), coupled with mass spectrometry, designed to identify potential “binding partners” of Kv1.2, followed by functional screening of the effects of the identified interacting proteins on the expression and biophysical properties of Kv1.2-encoded channels.

They report the identification of multiple Kv1.2-interacting proteins, one of which they went on to study in detail. This protein was Slc7a5, a neutral amino acid transporter. Experiments are presented demonstrating that co-expression of Slc7a5 with Kv1.2 reduces total Kv1.2 protein, and dramatically hyperpolarizes the voltage-dependence of activation of Kv1.2-encoded currents (by > 40 mV). They also show that the effects of Slc7a5 are attenuated with the additional co-expression of Slc3a2, a previously identified binding partner of Slc7a5. These latter observations were interpreted as reflecting Slc3a2-mediated reduced binding of Slc7a5 to Kv1.2, although this was not demonstrated directly. The reduction in Kv1.2-encoded currents with co-expression of Slc7a5 was attributed to an interesting and seemingly novel ‘inactivation trap’ mechanism.

In the final series of experiments, the authors go on to examine the effects of two recently reported variants of Slc7a5, linked to neurodevelopmental delay, and epilepsy-linked gain-of-function Kv1.2 mutants on Slc7a5-mediated modulation of heterologously expressed wild type and mutant Kv1.2-encoded currents. The work is carefully done and well-described. The data are of high quality and the manuscript is well-written.

Thank you for these positive comments about our manuscript. We would like to clarify that we did not intend to convey that we identified multiple Kv1.2-interacting proteins – we think it is more

accurate to say that we identified many candidates, but ruled out most of them by detailed screening. Slc7a5 (and indirectly, Slc3a2) were found to have dramatic gating effects on Kv1.2 that are far greater than other known accessory proteins of Kv1 channels, such as the canonical Kv β subunits. This was an unexpected finding that we feel highlights a gap in our understanding of Kv channels.

The authors might well be correct that the interaction between Slc7a5 and Kv1.2 and the observed functional effects of Kv1.2 currents “may influence neuronal function and disease pathogenesis in patients with Slc7a5 or Kv1.2 mutations”. Much more would need to be done, however, to demonstrate the physiological and/or the pathophysiological relevance/significance of the observations reported here. In the absence of experiments documenting neuronal expression of Slc7a5 and the functional effects of Slc7a5 co-expression on Kv1.2-homotetrameric or Kv1.2-heterotetrameric channels in neurons, the study represents little more than a well done heterologous co-(over)expression study in unknown physiological relevance/significance.

We agree that more work will be required to understand the physiological implications of our findings. In order to avoid ambiguity we have altered the wording of this part of the paper. The revised paper includes additional evidence documenting the presence of Slc7a5 protein and mRNA in cortical neurons (Fig. 6d,e), and expression of Slc7a5 and Kv1.2 in the same hippocampal and cortical neurons (Fig. 6f) from P0-2 rat pups. We have also added additional experimental details related to the mutual interactions of Kv1.2, Slc7a5, and Slc3a2, and the preserved gating effects of Slc7a5 in channels co-assembled with Kv β subunits.

Reviewer #2 (Remarks to the Author):

Summary:

Baronas and colleagues report that a previously unknown protein interaction between the Kv1.2 potassium channel and a neutral amino acid transporter (Slc7a5 / LAT-1) causes a marked suppression of current through a novel mechanism. The Kv1.2 / Slc7a5 interaction enhances C-type inactivation (faster, more complete) and produces an enormous hyperpolarizing shift (-47 mV) of activation such that, in combination, these effects result in an “inactivation trap” over a tested voltage range of -100 mV to -80 mV. It should be noted that the hyperpolarizing shift of activation alone is quite remarkable. I am not aware of any other channel subunit or cellular modification (e.g. phosphorylation, redox, lipid interaction) that is capable of causing such a large shift. The protein interaction is specific (e.g. current modification does not occur for Kv1.5; nor with Slc1a5 or Slc3a2 co-expression although the latter partially mitigates the Slc7a5 effect). Proximity of the Kv1.2 / Slc7a5 interaction is demonstrated with bioluminescence resonance energy transfer (BRET), and a “split” YFP reporter is used to show that the mitigating effect of Slc3a2 is not caused by displacement of Slc7a5. The final section of the paper highlights the potential significance of this new finding to human disease. Kv1.2 missense mutations associated with epileptic encephalopathy were previously reported to cause gain-of-function changes, which is difficult to reconcile with the clinical phenotype since a reduction of neuronal K conductance would be expected to be epileptogenic. The present study shows two different Kv1.2 mutant channels have exaggerated sensitivity to current suppression by Slc7a5, thereby implicating a possible loss-of-function pathogenesis. Conversely, missense mutations of Slc7a5 associated with recessively-inherited autism spectrum disorder greatly attenuated the suppression of

Kv1.2 current or in one case completely abolished the suppression.

Critique:

The remarkable findings of this study will be of high interest to biophysicists interested in regulation of ion channel gating and have far-reaching implications for the characterization and interpretation of disease-associated channelopathies, when studied in artificial expression systems. A number of questions need to be resolved (by comment or additional experiments) to more fully understand the potential biological implication of this novel protein-protein interaction.

Thank you for your positive comments on this work. We agree that the magnitude of the effects described in the paper are striking, and suggest a strong influence on channel gating generated by a unique combination of functional effects. We hope to clarify that we do not intend to imply that Slc7a5 is an obligate accessory protein of Kv1.2, and that the specific context of their association is not yet known. As shown later in the response (reviewer 2 comment 3), there is wide variability in terms of the relative expression levels of Kv1.2 and Slc7a5 in different cells, and so an obvious future goal is to test candidate cell types or conditions where this interaction may be more prominent.

1) The L-type 1 neutral amino acid transporter is a dimer of LAT1/CD98 (Slc7a5 / Slc3a2 gene products) cross-linked by a sulfhydryl bond. The profound effects on Kv1.2 gating were observed for LAT1 expressed alone, and were markedly attenuated by co-expression of CD98. In light of this, (a) would the (unbound) LAT1 protein be present in sufficient quantities to produce the gating effects on Kv1.2 channels in cells normally expressing the transporter, and (b) exposure to reducing agents (DTT or DTE) should be tested to see if this eliminates the mitigation of suppression attributed to Slc3a2 (CD98) co-expression.

We have included additional experimental findings that address the role of the disulfide bond between Slc3a2 and Slc7a5 in more detail. We could not use reducing agents to break this disulfide bond because Kv1.2 exhibits exquisite sensitivity to extracellular reducing conditions (we described this in a recent paper, Baronas et al. 2017, 'Extracellular redox sensitivity of Kv1.2 potassium channels') resulting in dramatic gating shifts (~70-80 mV), that would confound our observation of Slc7a5 effects. Instead, we tested the gating effects of Slc7a5 and Slc3a2 after mutation of cysteine residues responsible for the inter-subunit disulfide bond (Supplementary Fig. 1). Slc7a5[C164A] retains strong regulatory effects on Kv1.2, and these effects are rescued by transfection of sufficient Slc3a2. This is consistent with previous work mapping a broad hydrophobic surface contact between Slc3a2 and the closely related transporter Slc7a8 (LAT2) (Rosell et al., PNAS, 2015), suggesting many important contacts in addition to the disulfide bond. At this stage, we have not quantified the effects of this disulfide on the strength of the interaction. However, it is clear that it is not essential for the mutual modulation between Kv1.2, Slc7a5 and Slc3a2.

A related point is that a variety of Slc7 transporters are detected by RNAseq in neurons, which might also compete with Slc7a5 for interaction with Slc3a2, and it is not known how the pattern of expression of Slc3a2 and the various Slc7 transporters may vary on a cell-to-cell basis.

Lastly, it should be noted that the experimental conditions we presented were settled on to highlight the extremes of potential variability generated by Kv1.2, Slc7a5, and Slc3a2. In terms of transfection ratios we have typically used 2:1:2 (Kv1.2:Slc7a5:Slc3a2). Inclusion of more Slc7a5 (eg. 1:1 relative to Kv1.2) causes profound suppression of current and expression that make it very difficult to do experiments, and much weaker rescue is seen with Slc3a2. Inclusion of less Slc3a2 also results in less efficient rescue of the gating effects. This is mentioned on page 6 (related to Fig. 2a), and page 12 (related to transfection ratios in Fig. 7)

2) Related to item (1), the authors should test whether the profoundly enhanced suppression of current for the Kv1.2 mutations associated with epilepsy (Fig 9E, F) is prevented by co-transfection with Slc3a2. If co-expression of Slc3a2 largely prevents this loss-of-function effect, then the relevance to disease pathomechanism becomes uncertain.

We have added these experiments to the study (Supplementary Fig. 5). As mentioned, Slc7a5 effects depend on the relative amounts of Kv1.2, Slc7a5, and Slc3a2, which vary from cell to cell, and may also depend on the presence of other Slc7 transporter subunits. For example, using the Janelia NeuroSeq tool, the ratio of Slc3a2 and Slc7a5 varies by 5 orders of magnitude across different cell types categorized (ie. many cell types have a significant excess of Slc7a5 vs Slc3a2, at the mRNA level, and vice versa). The Kv1.2+Slc7a5 condition we have described should be considered as the ‘maximal effect’ seen with full stoichiometric assembly of Kv1.2 with Slc7a5.

In terms of the effects of Slc3a2, there is very wide cell-to-cell variability of the voltage-dependence of activation of the epilepsy-linked Kv1.2 mutants co-expressed with Slc7a5 and Slc3a2. We have added this data in Supplementary Figure 5. This variation is likely due to the vast dynamic range of voltage-dependence that can be sampled by the system, and likely variation in the expression of Slc7a5, Slc3a2 and Kv1.2 between cells. Importantly, some Slc7a5-mediated effects remain prominent in the Slc3a2 + Slc7a5 + Kv1.2(mutant) condition. For example, there is a clear deceleration of channel deactivation that persists after co-expression with Slc3a2 (Supplementary Figure 5c). Overall, this supports the notion that Slc3a2 can modulate Slc7a5 effects on the channel, but does not completely occlude Slc7a5 effects on Kv1.2. We bring this up along with the split-YFP experiment discussed below.

3) LAT1 / Slc7a5 is clearly expressed “in the brain”, but is it known whether expression occurs in neurons (with Kv1.2)? This is an important question since the highest expression in the brain is probably from endothelial cells to mediate amino acid transport across the blood brain barrier. If LAT1 is not in neurons, then the interesting findings in this study do not have much biological relevance (although they still identify and interesting biophysical effect on channels).

We detected Slc7a5 in cultures of dissociated rat neurons (enriched with cytosine arabinoside) using immunohistochemistry, western blot, and RT-PCR (Fig. 6). We have added these data although we are continuing to explore this in more detail. Like most ion channels, Slc7a5 transcript levels are not especially abundant in neurons. However, online resources including the Brain RNA seq database (https://web.stanford.edu/group/barres_lab/brain_rnaseq.html), and the Linnarson group (<http://linnarssonlab.org/cortex/> and most recently <http://mousebrain.org/geneseach.html>), indicate that Slc7a5 mRNA is present in neurons. Slc7a5 mRNA abundance in neurons is significantly

lower than in endothelial cells, but is comparable to Kv1.1, Kv1.2 and Kv β subunits in many cell types. Using the Janelia Neuroseq database we also generated the following scatterplots that map Kv1.2 and Slc7a5 transcript levels from single cell RNAseq:

These illustrate that many cells exhibit similar transcript levels of Slc7a5 and Kv1.2 (a similar plot was generated for Kv1.2 and Kv β 1 on the right, for comparison), although there is wide variability. Some cells express a substantial excess of Kv1.2, and some express an excess of Slc7a5.

4) What is the potential role of the KvBeta subunit in these phenomena? Do Itk- cells express KvBeta subunits? Does co-transfection of KvBeta alter the gating effects produced by Slc7a5? This is an important question because if the KvBeta subunit prevents the Slc7a5 suppression of current, then the biological relevance is again in question.

We added experiments demonstrating that Kv β subunits and Slc7a5 can simultaneously alter Kv1.2 gating (Fig. 4c-e). It is recognized that the mouse fibroblasts we use express an endogenous Kv β 2 subunit (but no detectable Kv currents), and we have detected Kv β 2 in mass spectrometry runs using Itk- fibroblasts to express channels. Therefore, we co-expressed Kv1.2 with Kv β 1.3 and Slc7a5 in HEK cells. These proteins generate currents that exhibit a combination of the dramatic leftward gating shift and hyperpolarization mediated disinhibition (due to Slc7a5), and N-type inactivation (due to Kv β association). This was an interesting addition to our study – similar to previous studies in BK channels demonstrating concomitant assembly of BK channels with β and γ subunits. We have included these data in a revised version of Fig. 4.

Specific Points:

1) Statistical tests of the perceived differences in current density (Fig 2a) are not provided. The authors are commended for showing the high variance of current density that occurs with transient transfections, but this also highlights the need for ANOVA.

Thank you for pointing this out. We have clarified statistical tests used here and elsewhere in the Figure legends. In this particular Figure, both Slc7a5 and the Slc7a5+Slc3a2 condition result in a statistically significant reduction of current density relative to WT Kv1.2 alone.

2) The effect of Slc7a5 co-transfection on Kv1.2 inactivation is a key contributor to the mechanism of the “inactivation trap” and understanding the relief of current suppression by hyperpolarization (Fig 3). As such, Supplementary Fig 2 showing the voltage dependence of inactivation should be moved to the main body of the paper. In addition, there may be a problem with the voltage protocol used to measure pre-pulse inactivation (Supplementary Fig 2). The inactivation curve for Slc7a5 (Fig S2) shows fully available channels (i.e. not inactivated) at about -80 mV, with a flat plateau to -110 mV. The data in Fig 3, however, show a large difference in available current produced by a holding potential of -120 mV versus -100 mV or -80 mV. This difference is not because of the Kv1.2[V381T] background used to promote C-type inactivation (Fig S2), because the data in Fig 5B (Kv1.2[V381T] channel) also show voltage-dependent relief of current suppression at -120 mV compared to -100 mV. So either the relief of suppression by hyperpolarization is not attributable to recovery from inactivation or the voltage-dependence of inactivation curve in Fig S2 is in error.

As suggested, we have moved the data illustrating Slc7a5 effects on voltage-dependence of inactivation to the main text (Fig. 5d).

The issue related to the voltage protocol is a challenging mechanistic aspect of our findings that is not yet clear to us. The voltage protocols are correct in terms of their description in the original manuscript. However, we have now clarified the steps involved in collecting the data. Upon achieving whole cell access, Kv1.2-mediated currents are initially very small, but can be disinhibited by hyperpolarization to negative voltages (~-120 mV), as shown in the original manuscript. After this disinhibition, we generate activation and inactivation curves, and current levels remain quite stable with a -100 mV holding potential. Recovery from inactivation induced during the steady-state inactivation protocol (currents generated by channels that have been disinhibited) occurs during the 7s interpulse interval (at -100 mV), suggesting that the rapid inactivation occurring during the pulse (ie. Fig. 5c) does not return channels to the same non-conducting state that they recover from with the initial -120 mV disinhibition protocol. Overall, this suggests that Slc7a5 promotes C-type inactivation of Kv1.2[V381T] but may also promote additional inactivated/non-conducting states during prolonged interaction with the channel. We have attempted to communicate this uncertainty more clearly (top of page 9).

3) The holding potential used for the data in Fig 2A should be stated, since the Slc7a5 suppression of Kv1.2 current is voltage-dependent.

We agree and we have clarified our description of this experiment. The current density was measured prior to disinhibition, with a holding potential of -80 mV, and a pulse to +60 mV (top of page 6)

Reviewer #3 (Remarks to the Author):

This paper reports two interesting new findings that are worthy of attention:

A. A functionally significant interaction between an amino acid transporter and a voltage-gated ion

channel.

- the full mass spec data should be provided as supplementary material or deposited in a public database.

We have included the full mass spec data as a supplementary file. We emphasize that due to the cross-linking approach we have taken, there is likely a large number of candidate proteins that are not interacting proteins of Kv1.2. We have made a note of this in the supplementary information.

B. The ability of co-expression of Slc7a5 to turn apparent gain-of-function-producing mutations into an effective loss of function. This is an interesting possible explanation for the otherwise puzzling observation that these mutations produce similar phenotypes in animals or humans to loss-of-function mutations.

However there are other aspects of the manuscript that are problematic and should be clarified or improved:

1. Is the electrophysiology done with the same 1.5/1.2 chimera that was used for the immunoprecipitation? If so, the abstract and text should reflect this. Also, some critical experiments would need to be repeated to support the case that native Kv1.2 participates in this same interaction.

We agree that this was not clear in the original version. We have now explicitly stated that the electrophysiological experiments were carried out using WT Kv1.2 channels (rat).

2. The use of the expression “inactivation trap mechanism” is misleading to channel biophysicists: it implies a mechanism involving retention of the Kv1.2 channels in the inactivated state. The association of Slc7a5 does not appear to function in this way; instead it promotes inactivation by favoring entry into the inactivated state, both by favoring entry into the open state and by speeding up open state inactivation. It would be better simply to say that it promotes inactivation. It might be even better to title the manuscript based on point #2 above.

We recognize that there is ambiguity in the use of this term in the title and abstract, without any prior context. Therefore, we have reworded these sections, and included a more complete discussion of the gating effects in the discussion section.

We would like to clarify that the combined effects of Slc7a5 (on activation and inactivation) cause stabilization in a non-conducting state that requires profound hyperpolarization for channels to recover (Fig. 3). These effects become especially pronounced with the large gating shifts seen with epilepsy-linked mutants, and the term ‘inactivation trap’ was intended to be a succinct term that captures this unique combination of effects on activation and inactivation.

3. The authors overstate several aspects of the story involving Slc3a2 rescue of the Slc7a5 phenotype.
- there is no basis for talking about the relative strength of the Kv1.2-Slc7a5 interaction relative to the Slc3a2-Slc7a5. There is no information about the amounts of proteins, only the amount of plasmid DNA.
- there is no indication that Slc3a2 co-expression produces a statistically significant reduction in Kv1.2 BRET with Slc7a5 (Fig. 6C; other pairs are marked with p-values but not that one; the data points shown

do not support a p-value < 0.05). If not, there is no place for a statement that it does.

We agree there was a lack of clarity in our presentation of these findings.

We have revised this section to remove mention of the relative strength of interaction. Related to this question there are a couple of noteworthy points. Firstly, we observed that titrating the amount of Slc3a2 plasmid leads to variable rescue of the gating effect of Slc7a5. Overall we found that a 2:1:2 (Kv1.2:7a5:3a2) ratio could fairly consistently rescue the Slc7a5 effect, and this is the condition that we characterized in detail. However, we feel it is important to recognize that the overall effect depends significantly on the relative amounts of each protein. For example, inclusion of more Slc7a5 leads to much stronger current suppression that is only weakly rescued. We have clarified that the substantial rescue by Slc3a2 depends on the experimental conditions (this is mentioned in the description of Figs. 2 and 7).

We also clarified the discussion of the findings and statistics in the BRET assay (page 10). I apologize that there seems to have been an error in the statistical test presented in the first draft, as the difference between the Slc3a2+Slc7a5 condition and the negative control was misreported. This condition is ambiguous as it is not statistically different from Slc7a5 alone, or the negative control, because the results are quite variable. The addition of additional data characterizing Slc3a2 rescue of Slc7a5 modulation of Kv1.2 mutants (Supplementary Figure 5) adds more information addressing the mutual interactions between these three proteins.

4. The split YFP experiments do not necessarily suggest “that Slc3a2 does not prevent assembly of Slc7a5 and Kv1.2.” If the membrane proteins associate more reversibly than the split YFP (and other results imply that this is so), the untagged Slc3a2 should lose out to the tagged Slc7a5, irrespective of their behavior without the split YFP.

We agree that we did not discuss this experiment with enough detail. We have rewritten this (page 11, 12) to emphasize that the flow cytometry experiments were complemented by electrophysiological experiments using all combinations of the split-YFP tags. Some of these details were not included originally in order to be concise.

An important finding now added to the revision is that split-YFP mediated fusion of Slc7a5 and Slc3a2 retains the ability to generate Kv1.2 gating shifts despite the promotion of the association of Slc7a5 and Slc3a2 by the split-YFP (Fig. 7d). Also, as presented previously, Slc3a2 can rescue current suppression of the Kv1.2-YFPC/Slc7a5-YFPN assembly (Fig. 7c), suggesting that Slc3a2 can influence the Slc7a5 effect on Kv1.2, even when the channel and transporter are held in close proximity. Overall, we agree that this complex will need further investigation, but the findings indicate that direct physical competition/occlusion is not likely the cause for attenuation of Slc7a5 effects by Slc3a2. This is also supported by the exaggerated effects of Slc7a5 on epilepsy-linked Kv1.2 mutants that persist even after co-expression of Slc3a2 (Supplementary Fig. 5).

It is also notable (we have clarified this in the revised text) that both Slc3a2 and Slc7a5 were ‘hits’ in our screen, but Slc3a2 was in fact a more confident ID. Slc7a5 was identified but with much smaller sequence coverage (fewer peptides, but quite abundant).

5. Mutants of Slc7a5 may affect folding or expression of the transporter protein. Without controls to test this, no conclusions should be made about the reduced effect on Kv1.2. It is in any case a stretch to imply that the disease consequences of these mutations to the changes in Kv1.2 rather than to the primary effect on amino acid transport.

We have expanded our discussion of these mutants. Past work on these mutants did not test their localization, but rather their transport activity in reconstituted vesicles (in the absence of Slc3a2), and they were reported to be transport deficient. Not much more is known about the molecular defects caused by these mutations. We confirmed their expression by western blot, now added to the Fig. 8d. We also failed to detect a BRET signal using either Slc7a5 mutant as an acceptor (Fig. 8b). Consistent with these findings, we have observed that expression of these mutants in cultured cortical neurons leads to a clear localization defect, as most of the transporters cluster in distinct puncta, in contrast to the relatively uniform expression of wild type Slc7a5 (Fig. 8e-g). These effects of the disease-linked Slc7a5 mutations have not been previously reported. In terms of the disease consequences of Slc7a5 mutations, we certainly agree that more work is required to distinguish amino acid transport effects vs ion channel gating effects. We have tried to be clear about this, but we also think it is important to keep an open mind about possible mechanisms of action. There are a variety of amino acid transporters with overlapping substrate specificity, and Slc7a5 may have multiple roles or multi-pronged effects that are important.

6. What is the authors' hypothesis to explain how Slc7a5 coexpression reduces the level of Kv1.2 protein (especially given that there is no change in the relative amount of surface expression)?

One possible reason for this effect is that there are ubiquitination sites (www.phosphosite.org) in the N-terminus of Slc7a5 that have been reported in mass spectrometry repositories. This is something that we will continue to explore. Nevertheless, there is a clear reduction in total Kv1.2 protein after co-expression with Slc7a5, in addition to the gating effects described, so we have made sure to clarify the dual nature of Slc7a5 effects on the ion channel.

7. Short of doing experiments in actual neurons, the authors could make a better case that these two proteins (Kv1.2 and Slc7a5) are likely to be co-expressed and associated in neurons. Transcriptomics data (Barres and Linnarsson publications) could help.

We have now included images and RT-PCR findings from dissociated hippocampal and cortical rat neurons illustrating the presence of Kv1.2 and Slc7a5 (Fig. 6). Previous histological work has suggested the presence of Slc7a5 at various locations in the rodent brain, and mRNA is detected using RNASeq approaches from neurons at levels comparable to Kv1.1 and Kv1.2. We have also included references to recent RNA seq studies and online tools. A valuable resource for us has been the Janelia NeuroSeq tool, along with a recent preprint from the Linnarsson group, which highlight the variation of Slc7a5 and Kv1.2 levels across a variety of neuronal cell types (mousebrain.org). Please see our response to Reviewer #2 (comment 3) for further discussion of this point.

8. The statement that attention to Kv1.2 as a source of structural information may have interfered with studies of its physiology has no basis and serves no purpose. It should be removed.

We have removed this particular sentence fragment and reworded accordingly. This was not meant to sound inflammatory although it probably came across that way.

Reviewers' Comments:

Reviewer #1:

Remarks to the Author:

The authors has responded in detail to the comments made in the critiques of the original submission, although they have not addressed the main issue raised about the unclear physiological relevance of the observations. In the absence of this, the paper, although very well done and clearly presented, provides detailed information on the modulatory effects of the Slc7a5 interacting protein on heterologously expressed Kv1.2-encoded channels, observations that may or may not be relevant to the functioning of Kv1.2-encoded channels in native cells....this important question has not been addressed.

Reviewer #2:

Remarks to the Author:

The authors have satisfactorily addressed my prior concerns. In particular, new data show (1) the Slc7a5 modulation of Kv1.2 persists when Kv-beta subunits are co-expressed, (2) the disulfide link between Slc7a4 and Slc3a2 is not required for the modulation of Kv1.2, (3) Slc7a5 is expressed in neurons as detected by immunohistochemistry, Western blot and RT-PCR. The new clarity on these three points strengthens the argument that Slc7a5 modulation of Kv1.2 likely occurs in the CNS in a physiological context.

Reviewer #3:

Remarks to the Author:

I am satisfied with the authors' thorough responses to all of the reviewer comments.

Reviewer #3 (Remarks to the Author):

I am satisfied with the authors' thorough responses to all of the reviewer comments.

Thank you for your input and efforts.